# Degradation & Restoration: A Low-cost Pipeline for Long-range Single-frame Turbulence Mitigation

## Abstract

Long-range turbulence mitigation (TM) remains challenging due to complex spatiotemporal distortions along the imaging path. Current approaches face several limitations in long-range TM: (i) traditional model-based image fusion methods fail to restore dynamic scenes, (ii) learning-based approaches demonstrate either inadequate distortion correction or poor deblurring performance, and (iii) simulators and synthetic training sets inadequately capture the characteristic features of long-range atmospheric turbulence. To achieve optimal restoration with minimal computation, we propose a low-cost single-frame TM pipeline featuring two key innovations: (i) a novel physically-grounded degradation simulator that enables rapid data generation while maintaining fidelity, and (ii) a simple yet effective parallel-training two-stage architecture for sequential distortion removal and deblurring. We demonstrate $4.3\times$ acceleration in degradation simulation and a minimum $2\times$ improvement in training efficiency compared to the baseline. Networks trained on our synthetic data consistently outperform those trained on other SOTA simulations. Our pipeline not only achieves state-of-the-art performance in single-frame TM but also surpasses many multi-frame approaches.

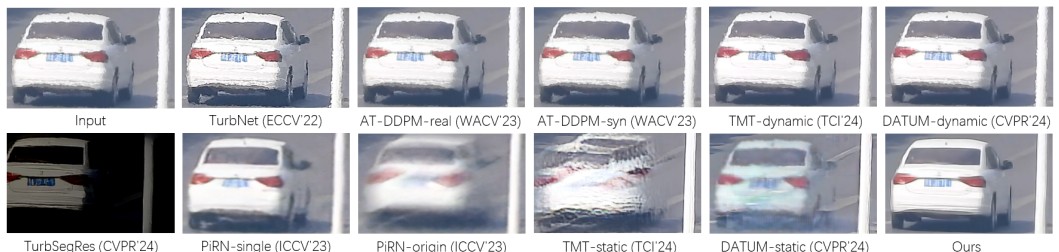

Figure 1: Real-world performance comparison on the RLR-AT benchmark (Xu et al., 2024).

## 1 Introduction

The presence of atmospheric turbulence leads to deformed shapes, blurred details, and reduced visibility, significantly affecting the performance of long-range observation such as face reconstruction (Nair et al., 2023; Jaiswal et al., 2023) and object recognition (Zhang & Chou, 2024; Deshmukh et al., 2013). Consequently, turbulence mitigation (TM) plays a critical role in long-range imaging systems by simultaneously enhancing visual quality and improving performance metrics for downstream tasks.

TM can be achieved through various approaches, each with inherent limitations. Under stable observation conditions, adaptive optics systems (Rao et al., 2016; Macintosh et al., 2006) can be deployed; however, these systems typically rely on large-aperture astronomical telescopes and require additional optical and electronic components, resulting in high costs and limited field deployability. Alternatively, traditional model-based multi-frame fusion methods offer a computational solution. However, these methods are computationally intensive and demonstrate effectiveness primarily for static scenes. As demonstrated in Figure 2, even state-of-the-art (SOTA) fusion methods (Lao et al., 2024; Xu et al., 2024) struggled to reconstruct moving objects.

Figure 2: Results of traditional pipelines on dynamic scene.

With the rapid advancement of deep learning, researchers have begun exploring learning-based image and video restoration approaches (Zamir et al., 2022; Liang et al., 2022) for TM (Mao et al., 2022; Zhang et al., 2024a). Although existing methods have shown promising performance on close-up real-world benchmarks such as Turbulence-Text (Mao et al., 2022), they exhibit limited restoration capability in the long-range benchmark RLR-AT (Xu et al., 2024). As shown in Figure 1, we compared recent SOTA methods (Mao et al., 2022; Nair et al., 2023; Jaiswal et al., 2023; Zhang et al., 2024b; Saha et al., 2024; Zhang et al., 2024a) in a dynamic scene and found that, regardless of the network architecture employed, the real turbulence images were not faithfully restored in terms of structural shape. This phenomenon indicates that the inability to recover the correct shapes is not primarily a deficiency of network architectures, but a consequence of shortcomings in the synthetic training data, most notably a mismatch in tilt simulation and an insufficient blur degradation domain.

Specifically, the displacement (tilt) fields produced by current tilt simulators deviate substantially from the spatial statistics of real long-range turbulence, and this mismatch directly undermines the faithful recovery of object shape. In addition, blur simulation also suffers from distinct limitations: Phase-to-Space (P2S) methods (Mao et al., 2021; Zhang et al., 2024a) generate degradation kernels via extensive Zernike coefficient computations, making them computationally expensive and difficult to scale, whereas the Gaussian-blur model of Saha et al. (2024) is computationally lightweight but produces only limited blur severity and variability, failing to cover the broader blur degradation manifold encountered in practice. Because models are not exposed during training to tilt and blur degradations that match the severity and statistical properties of real long-range turbulence, they inevitably fail to restore structural shapes under real conditions. In addition, the inherent slowness of existing simulators further impedes large-scale data generation and rapid model iteration.

During our effort to reproduce previous models, particularly DATUM (Zhang et al., 2024a) and TMT (Zhang et al., 2024b), we observe substantial computational resource demands for training. In our experiments, DATUM required cal50 days on dual NVIDIA A100 GPUs, while TMT required about 30 days. The extremely slow training convergence and high computational cost render model iteration based on these methods prohibitive.

To address these challenges, we propose a fast degradation simulator tailored for long-range turbulence and a low-cost restoration framework with strong generalization. Our main contributions are as follows:

- We introduce a multi-scale noise stacking scheme together with a random warp-time strategy to enrich pixel displacement fields for tilt simulation.

- We propose a lightweight yet effective random kernel generator to accelerate blur simulation. To enable smooth spatially varying convolution with discontinuous kernels, we further present a novel Mask-then-Conv strategy along with a fast mask-transition method.

- We present a parallel-training Detilt-then-Deblur architecture for single-frame TM, achieving an excellent balance between performance and efficiency.

- Our simulator delivers a 4.3× speedup over prior approaches while improving model generalization under real-world turbulence. Remarkably, our single-frame restoration framework reaches state-of-the-art performance with only 2.5 days of training on dual NVIDIA RTX A6000 GPUs, surpassing even some multi-frame counterparts.

## 2 RELATED WORKS

### 2.1 TURBULENCE SIMULATOR

Recent advances in turbulence simulation include the Phase-to-Space (P2S) simulators, as proposed by Mao et al. (2021); Zhang et al. (2024a). Instead of directly propagating phase screens through

split-step models, P2S extracts Zernike coefficients from randomly sampled phase aberrations, providing a compact representation of wavefront distortions. These coefficients are interpolated over the spatial grid to model local variations in turbulence strength. At each pixel location, the interpolated Zernike vector is transformed into a coefficient vector of basis kernels through the learned phase-to-space mapping. Consequently, the spatially varying blur process is reformulated as a weighted summation of a small set of basis convolutions, where each basis corresponds to a pre-computed blur kernel. This decomposition drastically reduces computational cost: rather than convolving every pixel with a unique PSF, the image is blurred by several spatially invariant kernels followed by per-pixel coefficient weighting.

However, the limitations of P2S are evident. First, computing Zernike coefficients is itself expensive, making the generation of degraded images relatively slow. Second, the blur representation is fundamentally constrained by the Zernike order: adopting higher-order expansions on large input images dramatically increases both GPU memory consumption and runtime, since more coefficients must be computed and the corresponding weighted summation across the image becomes heavier. Conversely, restricting the expansion to a low Zernike order reduces the computational load but narrows the blur domain, leading to insufficient modeling capacity and inaccurate reproduction of turbulence-induced blur.

To address these issues, a recent approach named QuickTurbSim (Saha et al., 2024) is introduced. It employs simplex noise in the pixel displacement technique to simulate tilt, and it also uses simplex noise to generate masks for different degrees of Gaussian blur. However, due to the one-hot nature of these masks, abrupt transitions occur between adjacent blur regions, resulting in visible artifacts that compromise visual fidelity. Moreover, its exclusive reliance on Gaussian kernels significantly reduces the generalization capability of models trained on the synthesized dataset, as this oversimplified approximation fails to capture the complex spatially varying blur characteristics inherent in real-world atmospheric turbulence.

## 2.2 SPATIAL-VARIOUS CONVOLUTION IMPLEMENTATION

In commercial imaging software such as Ansys Optics (formerly Zemax) (Nicholson, 2024), blur is simulated through spatially varying convolution, where PSFs are sampled on a grid and interpolated across the image domain before being convolved with the input. While this approach produces spatially varying blur consistent with physical optics, it involves repeated convolution operations at dense grid locations, which substantially slow down the simulation process.

## 3 PROPOSED METHODS

### 3.1 OVERVIEW

#### 3.1.1 DEGRADATION SIMULATOR

Our simulator is built on the conclusion of Chan (2022), that it is more appropriate to state the image formation model as

$$\boldsymbol{I}_{out} = B(T(\boldsymbol{I}_{in})), \tag{1}$$

where $\boldsymbol{I}_{in}$, $\boldsymbol{I}_{out}$ denotes the latent image and the simulated blurry image separately, $T(\cdot)$ the warping operator (tilt), and $B(\cdot)$ the blurring operator. Shown in Figure 3(a), the first stage gets a sharp image as input, then warps the image using the pixel displacement technique, which can simulate the tilt degradation of turbulence. The second stage employs a spatially varying convolution to introduce blur artifacts, thereby transforming the tilt output of Stage 1 into a degradation that better approximates the distribution of real turbulence, shown in Figure 3(b).

#### 3.1.2 PARALLEL-TRAINING TWO-STAGE TM ARCHITECTURE

As illustrated in Figure 3(c), we adopt a parallel-training Detilt-then-Deblur framework inspired by Zhang et al. (2024b), with a key modification: in the second stage, instead of incorporating the entire two-stage network and keeping the stage-1 module fixed, we train the stage-2 network independently using blur–GT image pairs. This innovation yields significant gains in computational efficiency through our training paradigm, which eliminates $N_2 + V \times M$ redundant forward passes of the stage-1 network, where $N_2$ represents training iterations, $V$ stands for validation frequency and $M$ indicates the cardinality of the validation set in stage-2 training. Our proposed framework

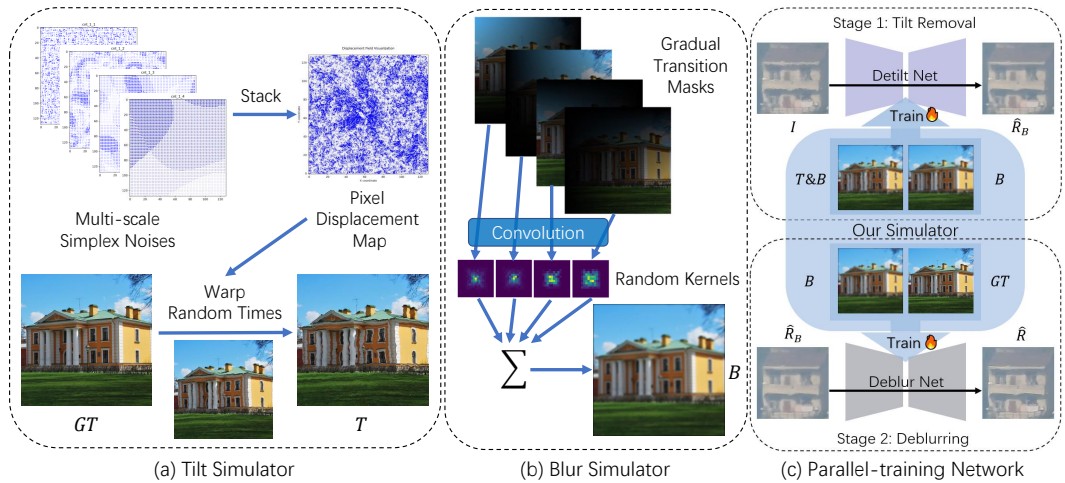

Figure 3: Main contributions of this paper.

enables parallel-training of the Detilt and Deblur networks, achieving at least 50% reduction in total training time compared to conventional sequential approaches.

## 3.2 TILT SIMULATOR

In our proposed tilt simulator, each warp operation is not limited to a single, mono-scale displacement. Instead, we model the displacement field as a hierarchical composition of layered 2D Simplex noise functions. Specifically, let the total number of noise layers be $L$. For the $i$-th layer, we denote the 2D noise function as $N_i(\cdot)$, with a progressive division factor $d$, and a weight $\alpha_p$ (controlled by the displacement factor together with noise persistence and lacunarity). Coarse-scale layers introduce global structural shifts, while fine-scale layers contribute localized fluctuations, ensuring that every displacement field captures both large-scale perturbations and subtle high-frequency distortions. The resulting horizontal and vertical displacement maps are defined as

$$\Delta x = \alpha_x \sum_{i=0}^{L-1} N_i^x\left(\tfrac{x}{d^i}, \tfrac{y}{d^i}\right), \quad \Delta y = \alpha_y \sum_{i=0}^{L-1} N_i^y\left(\tfrac{x}{d^i}, \tfrac{y}{d^i}\right), \tag{2}$$

where $(x, y)$ are the normalized pixel coordinates. The warped image is then obtained by resampling:

$$\boldsymbol{I}'(x, y) = \boldsymbol{I}\left(x + \Delta x, y + \Delta y\right), \tag{3}$$

with appropriate boundary clipping.

To further enhance realism, we do not apply a fixed number of warps. Instead, we introduce stochasticity into the number of pixel displacement iterations executed on each input, allowing different imaging instances to traverse atmospheric paths of varying lengths, with each segment subject to different turbulence strengths. Formally, if we denote a single turbulence-induced warp operator as $T(\cdot)$, then an input image after $t$ random warping iterations can be written as

$$\boldsymbol{I}_T = T^t(\boldsymbol{I}_{in}), \tag{4}$$

where $t$ is randomly sampled. By combining multi-scale displacement synthesis with randomized iterative warping, our simulator produces displacement maps of high richness and variability, which better approximate the stochastic nature of real atmospheric turbulence and improve the robustness of models trained under this simulation.

## 3.3 BLUR SIMULATOR

Since the strength of atmospheric turbulence varies spatially, the induced blur must also be modeled as spatially varying. Consequently, we aim to implement spatially varying convolution instead of

applying a fixed degradation kernel. A naive solution would be to assign a randomly generated kernel to each pixel, which is computationally inexpensive. However, this approach leads to excessive discrepancies between neighboring kernels, producing images that resemble pure noise instead of the desired smoothly varying blur.

Inspired by optical imaging software, we adopt the idea of sampling Point Spread Functions (PSFs) on a coarse grid and interpolating them across the spatial domain, followed by convolution to achieve spatially varying blur. Direct interpolation and per-pixel convolution, however, are prohibitively expensive on GPUs due to intensive memory access. To address this, we introduce a mask-based scheme in which gradual transition masks, combined with patch-level convolutions, replace the interpolation-plus-pointwise convolution pipeline. This design enables efficient GPU acceleration while preserving the smooth spatial transitions of turbulence-induced blur. The details of this implementation are as follows.

### 3.3.1 GRADUAL TRANSITION MASKS

To efficiently realize spatially varying blur, we design a mask generation strategy that replaces explicit PSF interpolation. The image plane is first partitioned into overlapping regions determined by a patch size parameter. For each region, we construct a radially decaying template that gradually decreases from the center toward the boundary. By sliding and overlapping these templates across the image, a complete set of smooth masks is generated.

At any spatial location, the contributions from neighboring regions overlap such that only four masks have non-zero values. These four masks are normalized to ensure that their sum equals one, and the local PSF at that location is computed as the weighted combination of the four corresponding regional PSFs. This guarantees two key properties: (1) Different regions contribute different kernels, enriching the variety of local blur patterns. (2) Because adjacent masks change gradually, neighboring pixels receive similar but not identical weights, producing seamless spatial variations rather than blocky discontinuities.

This mechanism achieves the same effect as the PSF interpolation strategy employed in Nicholson (2024), but in practice it is substantially faster and more memory-efficient, detailed in experiments.

### 3.3.2 MASK-THEN-CONV FRAMEWORK

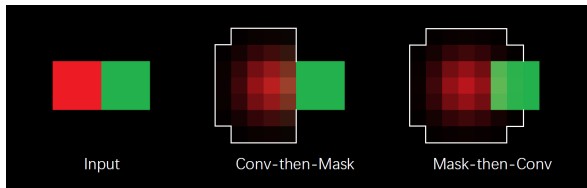

Figure 4: Comparison of Conv-then-Mask and Mask-then-Conv.

The distinction between the 'Mask-then-Conv' and 'Conv-then-Mask' strategies aligns with the broader discussion on product-convolution (scattering) versus convolution-product (gathering) formulations Chimitt et al. (2024). 'Conv-then-Mask' approach used in P2S and Saha et al. (2024) computes

$$I'(x, y) = \sum_k \big[ M_k \odot (I * K_k) \big](x, y),$$ (5)

where $I$ is the input image, $M_k$ denotes the binary or simplex-derived mask corresponding to the $k$-th kernel, and $*$ represents convolution. In this formulation, masking is applied after convolution, which corresponds to 'gathering' in Chimitt et al. (2024). This order inevitably introduces spatial discontinuities: regions near mask boundaries cannot accumulate sufficient kernel support, leading to incomplete blur formation and even chromatic distortions. In our case, this manifests as incomplete blur accumulation, spatial inconsistency, and chromatic discontinuities. In contrast, our proposed 'Mask-then-Conv' strategy adopts the product-convolution (scattering in Chimitt et al. (2024)) model, defined as

$$I'(x, y) = \sum_k \big[ (I \odot M_k) * K_k \big](x, y).$$ (6)

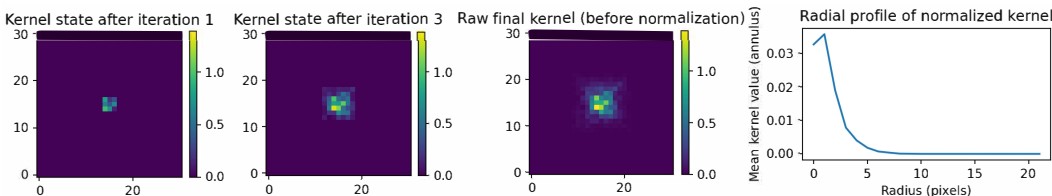

Figure 5: An example of our random kernel generator.

Here, the gradual transition masks talked above is performed first, and the masks are convolved with various kernels. This reordering ensures that every pixel fully inherits the convolutional support of the kernel while maintaining continuous transitions between regions. As a result, the synthesized blur fields preserve chromatic fidelity, avoid boundary artifacts, and more faithfully capture the gradual spatial variations inherent in turbulence-induced degradation, as Figure 4 shows.

### 3.3.3 RANDOM KERNEL GENERATOR

To better approximate real turbulence degradations, we first analyze measured PSFs and observe a characteristic energy distribution: a strong central peak with rapidly decaying surroundings, accompanied by non-uniform spatial asymmetry. Motivated by this, we propose a random kernel generation strategy.

Formally, let $\boldsymbol{K} \in \mathbb{R}^{s \times s}$ denote the kernel to be generated, and $c = \lfloor s/2 \rfloor$ the central index. We initialize

$$\boldsymbol{K} \leftarrow \mathrm{rand}(s, s) \cdot 10^{-4}. \tag{7}$$

Then, for each iteration $i = 1, \ldots, L$, we construct a random perturbation matrix

$$\boldsymbol{R}_i \sim \mathrm{rand}(2i + 1, 2i + 1), \tag{8}$$

and add it to the corresponding subregion of $\boldsymbol{K}$ centered at $(c, c)$, weighted by $\alpha_i$:

$$\boldsymbol{K}[c - i : c + i, \; c - i : c + i] \;\leftarrow\; \boldsymbol{K}[c - i : c + i, \; c - i : c + i] \;+\; \alpha_i \boldsymbol{R}_i. \tag{9}$$

The coefficient $\alpha_i$ decays step by step according to

$$\alpha_{i+1} \;=\; \alpha_i / \big(\mathrm{rand}() \cdot d_{\mathrm{diff}} + d_{\mathrm{min}}\big), \tag{10}$$

where $d_{\mathrm{min}}$ and $d_{\mathrm{diff}}$ control the decay rate. Finally, in order to maintain the overall luminance of the convolved image, we normalize $\boldsymbol{K}$. This procedure yields kernels with a pronounced central peak and rapidly decaying periphery, effectively mimicking the non-uniform energy distributions observed in real turbulence PSFs. Compared with prior methods, our random kernel generator maintains physical plausibility, introduces stochastic diversity, and enables efficient large-scale synthesis.

## 4 EXPERIMENTS

### 4.1 DATASETS AND EXPERIMENTAL SETTINGS

#### 4.1.1 TRAINING SETS

To ensure a fair comparison, we generated QuickTurbSim Saha et al. (2024) and our training sets with structure identical to the SOTA Zernike-based ATSyn-static dataset Zhang et al. (2024a) by randomly sampling parameters from the recommended ranges.

#### 4.1.2 EVALUATION BENCHMARK

We tested on the RLR-AT benchmark Xu et al. (2024). To the best of our knowledge, there are currently no authentic paired datasets for long-range TM in real-world scenarios. Our investigation reveals that RLR-AT stands as the only challenging real-world benchmark for long-range TM, as detailed in Appendix A.

Figure 6: Different backbones × different synthetic training sets.

### 4.1.3 IMPLEMENTATION DETAILS

All experiments were conducted on identical computing cluster nodes equipped with dual AMD 9004 series processors (64 cores) and an NVIDIA RTX A6000 graphics card. For fair comparison, regardless of the training set or the backbone architecture employed, each stage of the two-stage network underwent $4 \times 10^6$ iterations. The models were validated every $2 \times 10^5$ iteration. All remaining settings are consistent with the original OKNet Cui et al. (2024c) implementation. For evaluation, we selected the models that achieved the highest Peak Signal-to-Noise Ratio (PSNR) during validation. Unless otherwise specified, OKNet is used as the backbone.

## 4.2 DEGRADATION SIMULATOR

### 4.2.1 COMPARISON OF EFFECTIVENESS

Our experiments involved parallel training our two-stage network using three efficient image restoration backbones, CSNet (Cui et al., 2024a), ConvIR (Cui et al., 2024b), and OKNet (Cui et al., 2024c), on different training sets. The comparative visual results are presented in Figure 6, where each group of three images corresponds to the same backbone: the left image is trained on ATSyn (Zhang et al., 2024a), the middle on QuickTurbSim (Saha et al., 2024), and the right on our proposed dataset. The results demonstrate that across all three backbones, models trained on our dataset consistently achieve superior recovery of the van's geometric structure and substantially reduce boundary flickering artifacts compared to those trained on ATSyn or QuickTurbSim.

### 4.2.2 COMPARISON OF EFFICIENCY

We conducted a comparative analysis of computational time across different simulators on a dataset equivalent in scale to ATSyn-static, which contains 3000 GT images of $512 \times 512$ resolution, each paired with 50 degraded counterparts. Since the command-line interface of Zhang et al. (2024a) does not separately generate Tilt and Blur, we restricted the comparison to Turb image synthesis for fairness. All simulations used a batch size of 1. As summarized in Table 1a, our simulator achieves a $4.32\times$ acceleration over the previously fastest ATSyn simulator, while Chimitt & Chan (2020) is further limited to grayscale output, implying even lower efficiency for RGB synthesis. In addition, Figure 1b reports the runtime of different spatially varying convolution methods on $1280 \times 720$ images with $16 \times 9$ grids, where our method significantly outperforms Nicholson (2024).

| Algorithm | Device | Time (h)↓ |
|---|---|---|
| Chimitt & Chan (2020) | CPU | 43.11 |
| Saha et al. (2024) | CPU | 24.24 |
| Mao et al. (2021) | GPU | 17.85 |
| Zhang et al. (2024a) | GPU | 5.96 |
| Ours | GPU | 1.38 |

(a) Simulation time of different simulators.

| Algorithm | Time (s)↓ |
|---|---|
| Nicholson (2024) per line | 2.7041 |
| Nicholson (2024) per 16 lines | 1.4614 |
| Ours | 0.1715 |

(b) Time comparison of spatial various convolution.

Table 1: Overall comparison of simulation efficiency.

### 4.2.3 ABLATION STUDY

Figure 7 shows a comparison on RLR-AT benchmark. (a) Replacement of our Mask-then-Conv strategy with QuickTurbSim Saha et al. (2024) Conv-then-Mask approach; (b) Substitution of our randomized multi-warp scheme with single-warp degradation; (c) Exchange of our random kernel generation by QuickTurbSim's Gaussian kernels; (d) Replacement of our spatially-varying convolution with uniform kernel application. Visual results demonstrate that:

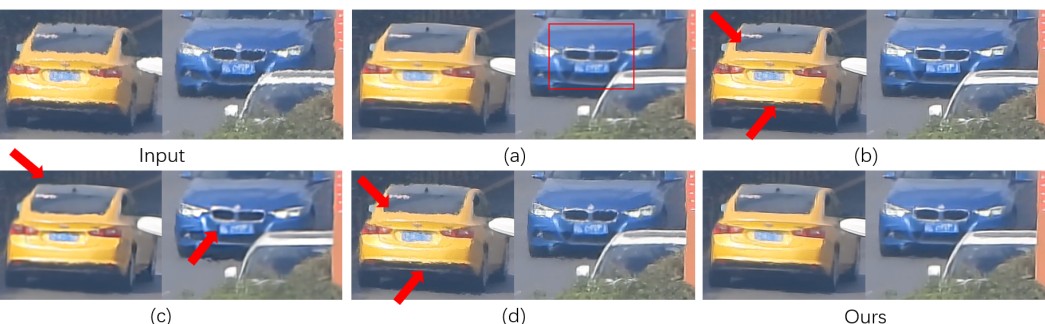

Figure 7: Ablation study of degradation simulator.

- The Conv-then-Mask approach (a) slightly impacts the TM model's capacity to learn distortion patterns characteristic of atmospheric turbulence degradation, resulting in asymmetric restoration of the blue car's front grill (highlighted in red box).

- The single-warp degradation (b), various Gaussian kernels (c), and uniform convolution (d) exhibit varying degrees of shape restoration failure, with (c) additionally introducing severe blurring artifacts (pointed by red arrows). These comparisons validate the necessity of each proposed component in our degradation simulator.

### 4.3 PARALLEL-TRAINED TWO-STAGE TM ARCHITECTURE

#### 4.3.1 COMPARISON OF REAL-WORLD TM VISUAL RESULT

We compared our method with the SOTA TM methods TurbNet Mao et al. (2022), PiRN Jaiswal et al. (2023), AT-DDPM Nair et al. (2023), TMT Zhang et al. (2024b), DATUM Zhang et al. (2024a) and TurbSegRes Saha et al. (2024) as shown in Figure 1 for dynamic scene and in Figure 8 for the static scene. Additional visual examples are provided in Appendix C. All baseline methods are evaluated using their official implementations. It should be noted that (i) the input patch size is set to $512 \times 512$ when testing AT-DDPM, TMT and DATUM due to memory restriction. (ii) PiRN-single uses only a single image as input, while PiRN-origin uses weighted averaging ($\frac{2I_t+I_{t+1}+I_{t+2}+I_{t+3}+I_{t+4}}{6}$) as specified in their source code. For the static scene, we compare only against single-frame baselines and additionally include DATUM-static as a multi-frame reference. For both dynamic and static scenes, our method produces the most accurate shape restoration while maintaining natural contrast throughout the entire scene.

#### 4.3.2 EVALUATION ON AN OCR BENCHMARK UNDER TURBULENCE

We further evaluate our method on a widely used OCR benchmark under turbulence Mao et al. (2022), following the evaluation protocol of Mao et al. (2022); Jaiswal et al. (2023). We adopt PaddleOCR Cui et al. (2025) with its latest PP-OCRv5 model and use only the text detection and recognition modules, excluding the unwarping module, to assess how well turbulence mitigation preserves information critical for downstream perception. For qualitative analysis, we follow Mao et al. (2022); Jaiswal et al. (2023) and report AWDR and AD-LCS. The results are shown in Table 2. For both AWDR and AD-LCS, our method consistently achieves the best performance across all metrics, establishing new state-of-the-art results on this benchmark.

| Algorithm | AWDR↑ | AD-LCS↑ |
|---|---|---|
| TurbNet Mao et al. (2022) | 0.966 | 3.744 |
| PiRN-single Jaiswal et al. (2023) | **0.998** | 3.290 |
| AT-DDPM-real Nair et al. (2023) | 0.902 | 0.022 |
| AT-DDPM-syn Nair et al. (2023) | 0.964 | 0.030 |
| **Ours** | **0.998** | **4.146** |

Table 2: Quantitative results on single-frame methods.

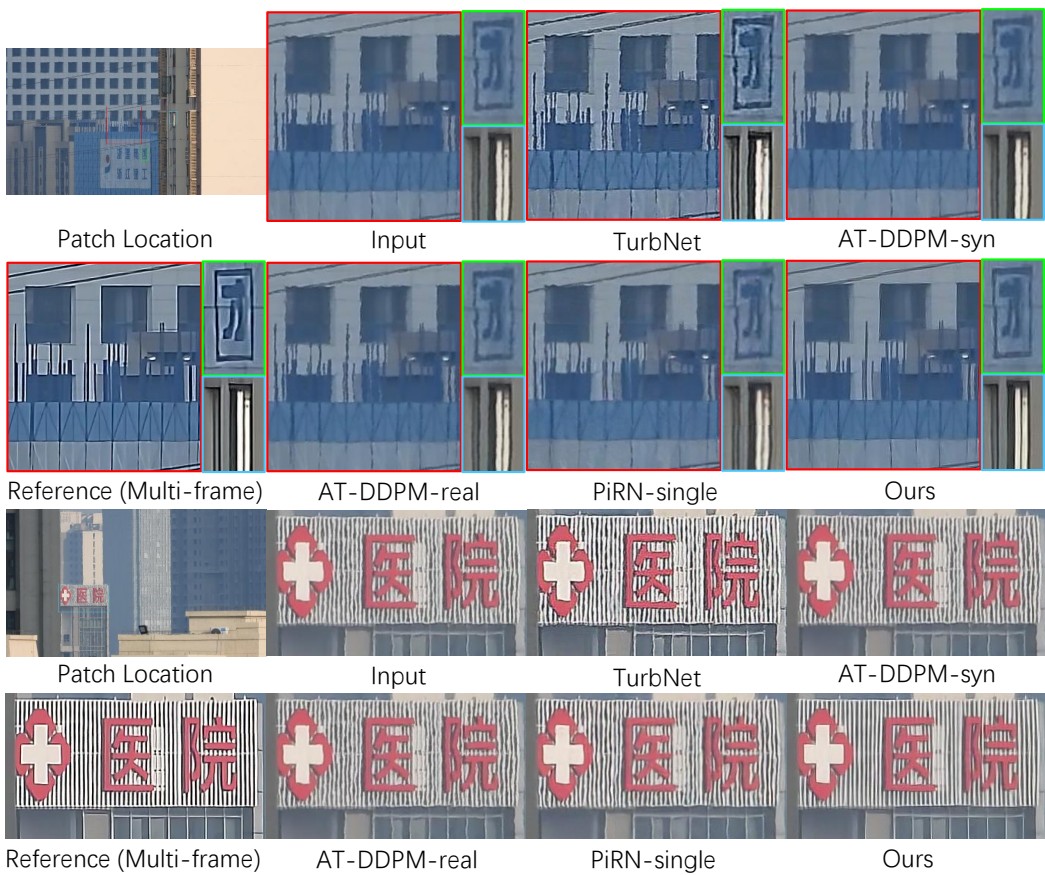

Figure 8: Comparison of single-frame TM methods on static scene.

### 4.3.3 COMPARISON OF TEMPORAL STABILITY

We compare temporal slices in Figure 10, where each column represents a frame cut from the same position in the image sequence. This experimental protocol was conducted following Xu et al. (2024); Cai et al. (2024), which can evaluate temporal distortions caused by spatio-temporal turbulence Cai et al. (2024): The temporal smoothness of pixel-wise tracking trajectories demonstrates a strong positive correlation with the model's capacity for temporal distortion restoration.

- In comparisons among single-frame methods (marked yellow), our approach demonstrates superior temporal smoothness and the most robust capability for temporal distortion restoration.

- When evaluated against multi-frame methods (marked blue), our method outperforms DATUM Zhang et al. (2024a) while achieving performance comparable to TMT Zhang et al. (2024b) and TurbSegRes Saha et al. (2024) in partial pixel tracking, pointed by red arrows. This represents a particularly notable achievement given that our model is trained on single-frame inputs without explicit learning of the temporal turbulence distribution. The observed performance indicates that our simulator effectively captures the spatial characteristics of turbulence degradation, while our model successfully learns these patterns, thus compensating for the absence of temporal modeling.

- Compared with networks trained in the QuickTurbSim and ATSyn datasets (unmarked), our line represents the smoothest, providing further validation that our simulator better enables networks to learn authentic turbulence removal.

### 4.3.4 OBJECT DETECTION ON RESTORED IMAGES

Figure 11 shows comparisons of object detection in restored images using widely used YOLO11x Jocher et al. (2023). The threshold was deliberately set to $50\%$ to demonstrate more effectively the superior performance of our approach, while other settings remained as default. The experimental

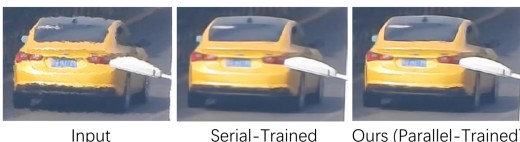

Input     Serial-Trained     Ours (Parallel-Trained)

Figure 9: Ablation study of training strategy.

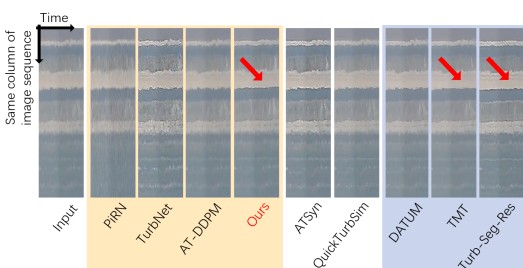

Figure 10: Comparison of time scale stability.

Figure 11: Detection comparison using YOLO11x Jocher et al. (2023).

results clearly demonstrate superior recognition confidence in our restored images, with all three objects above the threshold. We provide uncropped images for testing in the supplementary material.

### 4.3.5 ABLATION STUDY

Table 3 compares PSNR and SSIM Wang et al. (2004) in the Test Set with different training strategies. The notation X-Y in Stage-1/2 indicates using X as input and Y as target during training. 'continue' in Stage-2 denotes jointly training both stages while fixing Stage-1 weights. The results show that our training strategy performs slightly lower than TMT Zhang et al. (2024b) on synthetic data within acceptable margins, while achieving over $2\times$ training speed gains. Experimental results using Turb-Tilt in the first stage demonstrate the genuine efficacy of the Detilt-then-Deblur architecture.

| Stage-1 | Stage-2 | PSNR↑ | SSIM↑ |
|---------|---------|-------|-------|
| Test Set Turb | | 24.20 | 0.6930 |
| Turb-Blur | continue | 26.85 | 0.9101 |
| Turb-Tilt | Tilt-GT | 24.85 | 0.8581 |
| Turb-Blur | Blur-GT | 26.45 | 0.9062 |

Table 3: Comparison of training strategy.

Figure 9 compares the visual result between serial-trained and our parallel-trained network. Under real-world turbulence conditions, while serial-trained achieves superior shape reconstruction, it produces noticeably blurred outputs. In contrast, our method demonstrates stronger deblurring performance while maintaining an acceptable shape.

Figure 6 not only shows the effectiveness of our simulator, but also shows the universality of our parallel-training two-stage pipeline. Crucially, the proposed framework consistently achieves optimal shape and sharpness regardless of the IR backbone employed.

## 5 CONCLUSION

In this paper, we introduce a low-cost pipeline for long-range TM, including a well-designed fast degradation simulator and a simple yet effective parallel-training two-stage TM architecture. In the field of turbulence degradation, we contribute a enhanced tilt simulator using multi-scale noise stacking technique, an accelerated approach to perform smooth spatially-varying convolution and a new random kernel generator. Our simulator achieves SOTA performance both in degradation modeling capability and computational efficiency. Whether on visual quality or downstream task, our workflow achieves SOTA on single-frame approaches and outperforms most of the multi-frame methods. Our simulator and pre-trained models will be open-sourced upon publication.

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

## A  OTHER RELATED WORKS

### A.1  LEARNING-BASED TM METHODS

#### A.1.1  MULTI-FRAME APPROACHES

Zhang et al. (2024a) introduced a deep learning-enhanced pipeline that retains the traditional framework while replacing conventional modules with neural networks. Meanwhile, Zhang et al. (2024b) proposed a two-stage model that decouples tilt correction and blur removal, providing a modular architecture that inspired our design. Despite their effectiveness, these multi-frame methods often require substantial computational resources, mentioned in the Introduction. Additionally, these models are typically specialized for static or dynamic scenarios, limiting their generalizability. Building on Zhang et al. (2024a), Cai et al. (2024) exploited the low-pass filtering characteristics of regularized temporal representations to enhance temporal coherence in turbulent video sequences. A distinct approach was taken by Saha et al. (2024), who segmented moving objects from static backgrounds, restored them independently, and combined the results additively, demonstrating improved adaptability in dynamic scenes.

#### A.1.2  SINGLE-FRAME APPROACHES

Jaiswal et al. (2023) developed a two-stage framework, first employing a physics-integrated restoration network and then refining outputs via diffusion-based stochastic enhancement to improve perceptual quality metrics. Similarly, Nair et al. (2023) applied diffusion models to reconstruct human faces degraded by atmospheric turbulence. Mao et al. (2022) proposed a U-Net architecture with a redegradation module to enhance robustness. However, as noted in Xu et al. (2024), existing methods exhibit critical limitations: Jaiswal et al. (2023) struggles with deblurring, while Mao et al. (2022) underperforms in tilt correction —— highlighting the demand for more versatile and adaptive solutions.

#### A.1.3  EFFICIENT IMAGE RESTORATION METHODS USED IN OUR EXPERIMENTS

To validate the effectiveness and generalizability of our simulator and parallel-training strategy, we employed various efficient Image Restoration (IR) networks OKNet (Cui et al., 2024c), CSNet (Cui et al., 2024a), and ConvIR (Cui et al., 2024b) as backbones.

### A.2  REAL-WORLD BENCHMARKS

Table 4 shows real-world benchmarks mostly used in previous studies. Among all benchmarks, the RLR-AT Xu et al. (2024) has the highest resolution and the longest shooting distance, posing significant challenges for turbulence mitigation. As illustrated in Figure 1 of our paper, none of the previous methods can effectively restore it, which is precisely why we selected it as our benchmark.

The RLR-AT benchmark was captured using a Nikon Coolpix P1000 camera at 3000mm equivalent focal length and 30fps, recording turbulence-degraded scenes at distances of 1-13 km. It comprises 60 dynamic scenes and hundreds of static scenes. Since static scenes in RLR-AT benchmark can already be well restored using CDSN Xu et al. (2024), our evaluation focuses mainly on dynamic scenes in this benchmark.

## B  SIMULATOR DETAILS

### B.1  TILT SIMULATOR

As illustrated in Figure 12, the first row demonstrates a single warp process. By progressively injecting multi-scale noise, the displacement field evolves from subtle, nearly uniform perturbations into spatially varying shifts with heterogeneous magnitudes, resembling the random fluctuations observed in turbulence. The noise scale and the number of stacked noise layers are carefully controlled to introduce diversity across both spatial scales and local details. As the warp iterations accumulate, the displacement field gradually exhibits chaotic patterns, akin to the cumulative distortions in real

| Name | Distance↑ | Resolution | Subject | Note |
|---|---|---|---|---|
| Heat Chamber Mao et al. (2022) | $20m$ | $440 \times 440$ | static | gas heat; with GT |
| Turbulence Text Mao et al. (2022) | $300m$ | $440 \times 440$ | static | text only |
| OTIS Gilles & Ferrante (2017) | $\leq 1km$ | $\leq 520 \times 520$ | both | 3 static & 4 dynamic |
| BRIAR Cornett et al. (2023) | $\leq 1km$ | Various | dynamic | private |
| DOST (URG-T) Qin et al. (2024) | $\leq 1km$ mostly | $1920 \times 1080$ | dynamic | - |
| CLEAR Anantrasirichai et al. (2013) | $\leq 2km$ | $\leq 512 \times 512$ | dynamic | 3 scenes |
| TSR-WGAN Jin et al. (2021) | $\leq 3km$ | $\leq 1144 \times 744$ | dynamic | AVI compressed |
| TurbRecon Mao et al. (2020) | $\leq 4km$ | $512 \times 512$ | static | gray scale; 4 scenes |
| RLR-AT Xu et al. (2024) | $1km \sim 13km$ | $1920 \times 1080$ | both | - |

Table 4: Real Datasets and Benchmarks.

turbulence. To preserve a rich range of displacement intensities, we regulate the initial scaling factor along with the number of warps.

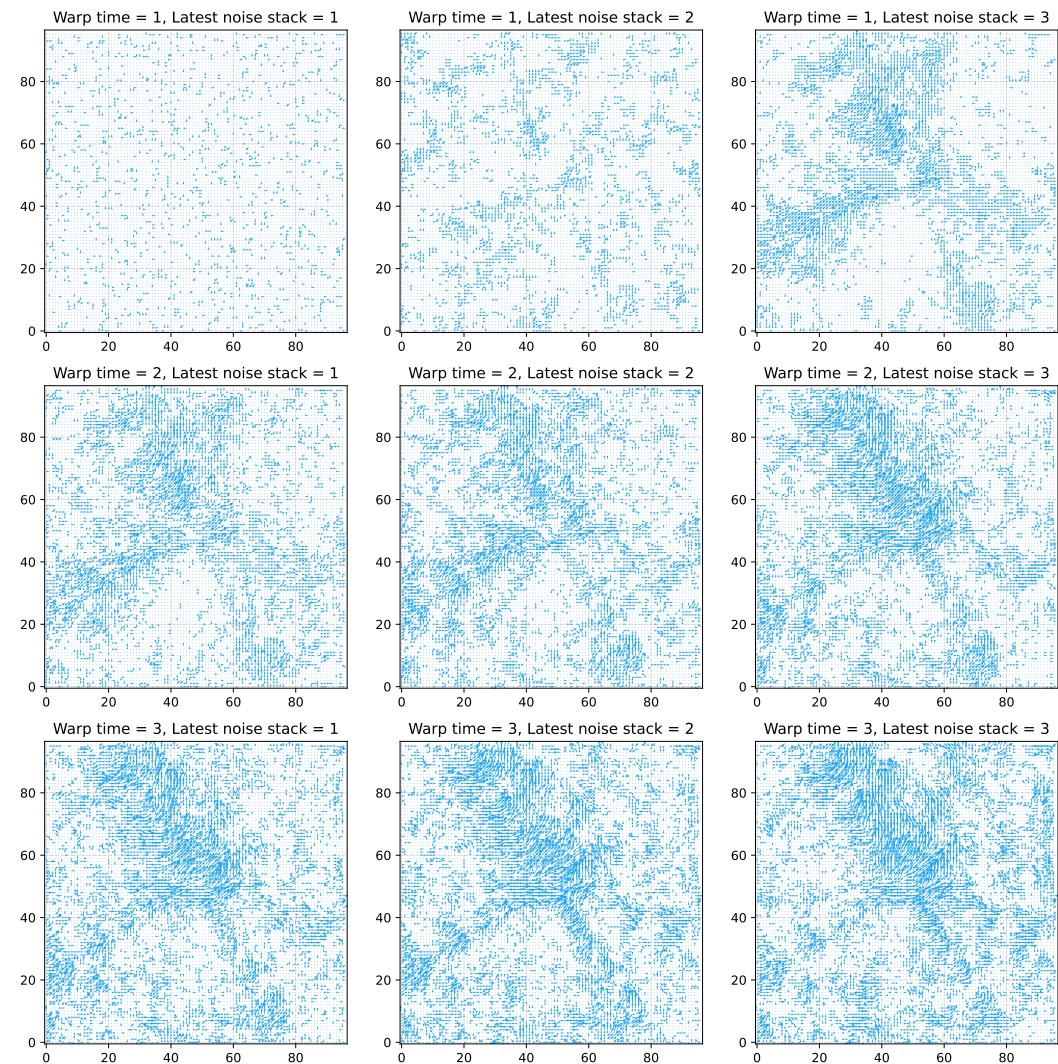

Figure 12: An example of displacement field generation.

## B.2 BLUR SIMULATOR

Due to space constraints, we describe the gradual transition mask and the random kernel generator only through textual explanations and simplified formulations. The complete algorithms and implementation details are provided in Algorithm 1 and 2. We further provide a comparison between the one-hot mask of Saha et al. (2024) and our proposed gradual transition mask in 13, facilitating a clearer understanding of the differences in mask design. And we also provide a comparison of kernels generated by different simulators, shown in 14.

## C MORE EXPERIMENTS

### C.1 SIMULATOR COMPARISON

Figure 15 shows the visual result of each simulator, Chimitt & Chan (2020), P2S Mao et al. (2021), ATSynZhang et al. (2024a) QuickTurbSim Saha et al. (2024) and ours. It should be specifically

**Algorithm 1** Generate Masks

**Input**: $w, h, size_p$
**Output**: $frames_{masked}$

1: $num_x \leftarrow (w-2)/size_p + 2$
2: $num_y \leftarrow (h-2)/size_p + 2$
3: $line \leftarrow [1, 2, ..., size_p, size_p - 1, ..., 1]$
4: $template \leftarrow line^T \times line/size_p^2$
5: $frames = zeros([(num_x + 1) * size_p - 1, (num_y + 1) * size_p - 1, num_x * num_y])$
6: **for** $i \leftarrow 0, num_x$ **do**
7:    **for** $j \leftarrow 0, num_y$ **do**
8:       $frames[i*size_p : (i+2)*size_p-1, j*size_p : (j+2)*size_p-1, i*num_y+j] \leftarrow template$
9:    **end for**
10: **end for**
11: **return** $frames[size_p - 1 : w + size_p - 1, size_p - 1 : h + size_p - 1, :]$

**Algorithm 2** Generate Random Kernel

**Input**: $size, level, start, div_{min}, div_{diff}$
**Output**: $kernel$

1: $kernel \leftarrow rand([size, size]) * 0.0001$
2: $num \leftarrow 1.$
3: **for** $i \leftarrow start, level - 1$ **do**
4:    $kernel[size/2 - i : size/2 + i + 1, size/2 - i : size/2 + i + 1] += rand([2 * i + 1, 2 * i + 1]) * num$
5:    $num/ = rand() * div_{diff} + div_{min}$
6: **end for**
7: $kernel/ = sum(kernel)$
8: **return** $kernel$

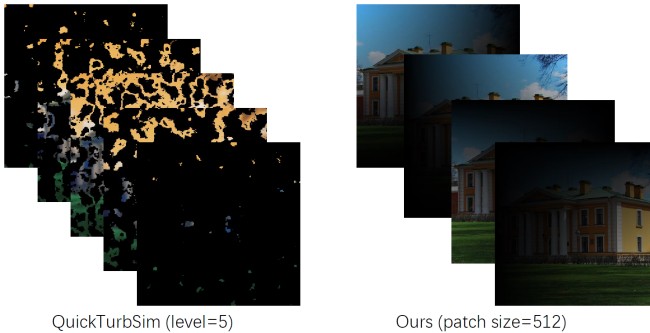

QuickTurbSim (level=5)       Ours (patch size=512)

Figure 13: Mask shape comparison.

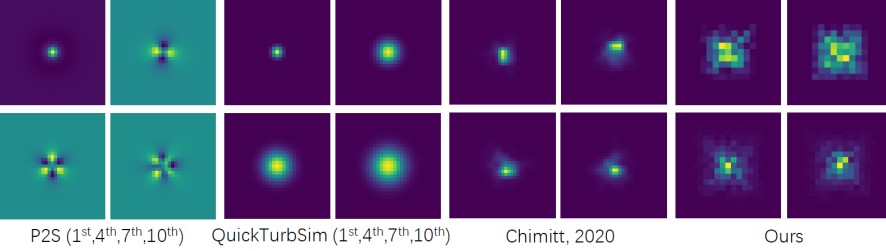

P2S ($1^{st}, 4^{th}, 7^{th}, 10^{th}$)    QuickTurbSim ($1^{st}, 4^{th}, 7^{th}, 10^{th}$)    Chimitt, 2020      Ours

Figure 14: Kernels comparison.

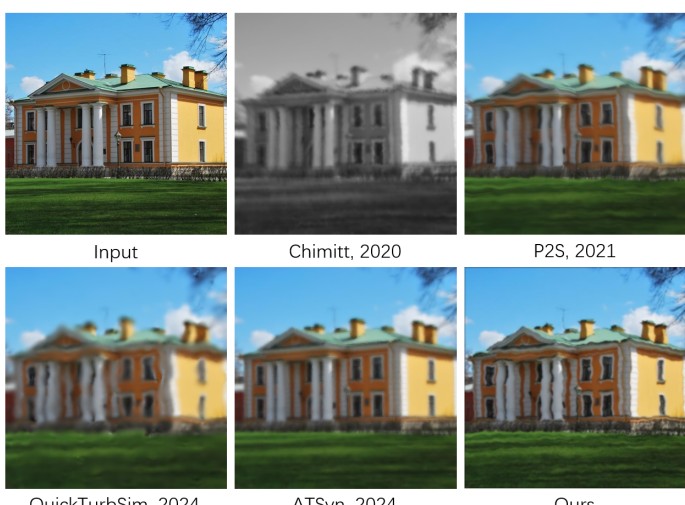

Figure 15: Simulator Comparison.

noted that, as each simulator employs distinct parameter configuration methods, we cannot use this figure to evaluate the relative merits of different simulators. The figures are provided only for illustrative purposes.

## C.2   MORE REAL-WORLD VISUAL RESULTS

We conducted additional evaluations on more static-scene data provided in Xu et al. (2024). We compared several single-frame methods TurbNet Mao et al. (2022), PiRN Jaiswal et al. (2023) and AT-DDPM Nair et al. (2023) and used DATUM-static Zhang et al. (2024a) as the reference. As shown in Fig. 16, our method consistently reconstructs object shapes more accurately across all scenes and better preserves the original brightness (whereas both TurbNet and the reference fail to maintain the correct brightness).

## C.3   GENERALIZATION STUDY ON SIMULATED DATASETS

Due to the physical properties of real long-range turbulence, acquiring paired clean ground truth is extremely difficult. To the best of our knowledge, no real-world long-range turbulence dataset provides paired clean and degraded images. Therefore, reference-based IQA metrics such as PSNR, SSIM, and LPIPS **cannot be used**.

Because different works adopt different turbulence simulation strategies, evaluating existing models on our simulated data, or evaluating our model on previously released synthetic datasets, mainly reflects (1) the coverage of each simulated distribution and (2) the model's ability to generalize across different types of synthetic degradation. It does *not* reflect absolute turbulence mitigation capability.

To ensure fairness, we train the *same* model on three synthetic datasets: QuickTurbSim Saha et al. (2024), ATSyn Zhang et al. (2024a), and ours. We evaluate PSNR on the corresponding test sets. The results are summarized below.

| Input PSNR | Trained on ATSyn Zhang et al. (2024a) | Trained on Ours |
|---|---|---|
| 22.81 | 22.66 (-0.15) | **23.57 (+0.76)** |

Table 5: Evaluation on QuickTurbSim. Models trained on our dataset generalize better to Quick-TurbSim's distribution.

Table 5 and Table 6 demonstrate that models trained on our simulated data generalize better.

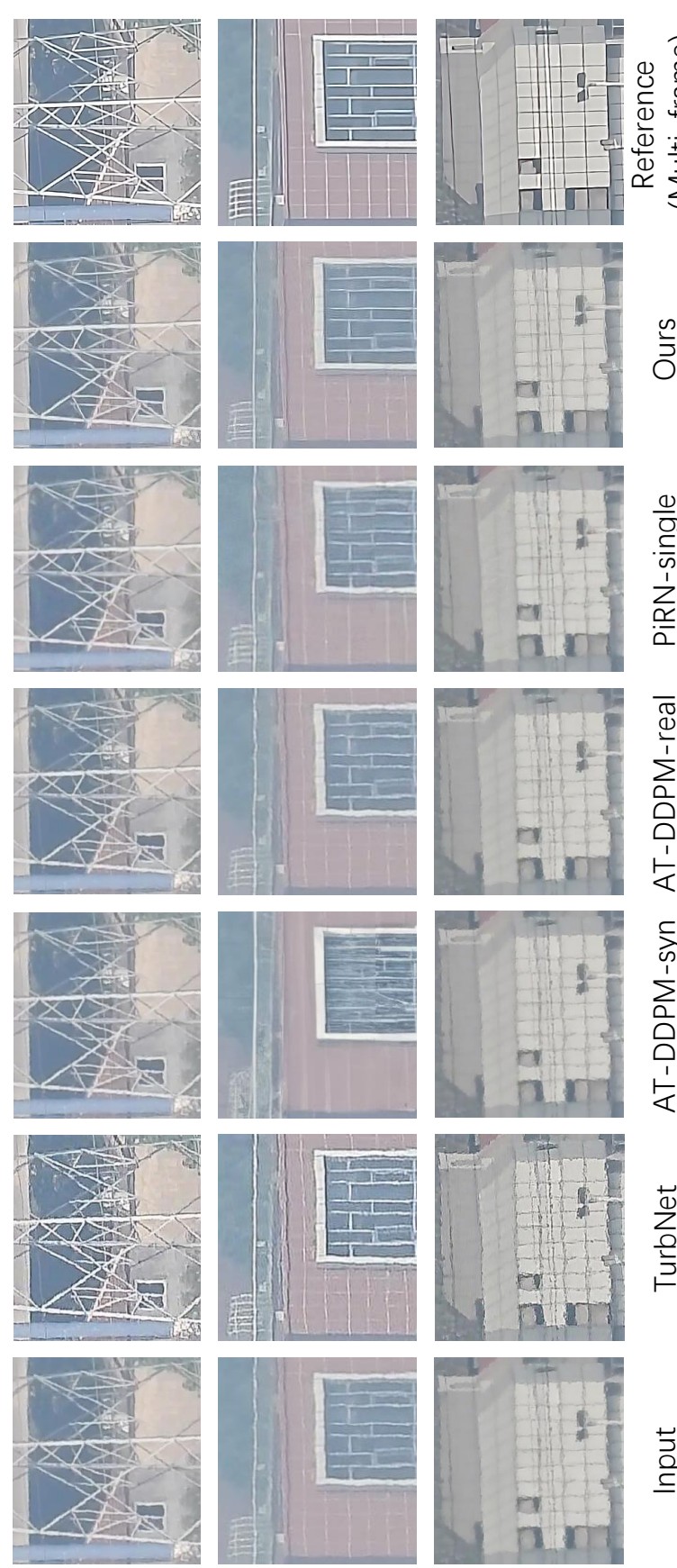

Figure 16: More Real-world static scenes

| Input PSNR | Trained on QuickTurbSim Saha et al. (2024) | Trained on Ours |
|:---:|:---:|:---:|
| 21.36 | 21.12 (-0.24) | **21.49 (+0.13)** |

Table 6: Evaluation on ATSyn. Models trained on our dataset again show stronger generalization.

| Input PSNR | Trained on QuickTurbSim | Trained on ATSyn | Trained on Ours |
|:---:|:---:|:---:|:---:|
| 22.92 | 22.73 (-0.19) | 22.44 (-0.48) | **26.41 (+3.49)** |

Table 7: Evaluation on our long-range synthetic turbulence data. Models trained on other datasets fail to generalize effectively.

Table 7 shows that models trained on previously released synthetic datasets do *not* generalize to our long-range turbulence degradation. This also indirectly explains why models trained with previous simulators perform worse than ours when applied to real long-range turbulence scenarios.

Note: We are unable to host the original 500+GB of data online due to storage cost limitations. For fairness, the test sets of QuickTurbSim Saha et al. (2024) and our simulated data were regenerated using the same script and identical settings.

### C.4 OBJECT DETECTION

In the supplementary material, we include two folders:

- In the folder named 'figure_11', we provide the original images used in Figure 11 of our paper in subfolder 'outputs', along with our object detection code 'test.py'. After running 'test.py', you can find results in the path 'detection/1'.

- In the folder named 'train', we provide three video clips that demonstrate detection performance in an additional 'train' scene. The experiment settings are the same as those mentioned in our paper.
  - '1_versus_SOTA_simulators.mp4' shows a comparison of the same network trained on different synthetic training set generated by different simulators;
  - '2_versus_SOTA_single_frame_methods.mp4' shows a comparison between single-frame State-Of-The-Art (SOTA) approaches;
  - '3_versus_SOTA_multi_frame_methods.mp4' shows a comparison of SOTA multi-frame approaches and our single-frame approach.

Both results demonstrate that our turbulence mitigation pipeline significantly enhances the detection capability using YOLO11x Jocher et al. (2023), which also confirms that our network can restore superior visual quality.

Previous state-of-the-art methods compared with:

- Simulators: ATSyn Zhang et al. (2024a), QuickTurbSim Saha et al. (2024);
- Single-frame methods: TurbNet Mao et al. (2022), PiRN Jaiswal et al. (2023), AT-DDPM Nair et al. (2023);
- Multi-frame methods: TMT Zhang et al. (2024b), DATUM Zhang et al. (2024a), Turb-Seg-Res Saha et al. (2024).

## D FUTURE WORKS

In this work, we proposed a fast simulator tailored for single-frame restoration and introduced a low-cost single-frame restoration method. While the trained models already surpass certain multi-frame approaches in terms of structural fidelity, the intrinsic limitation of single-frame information remains evident. Multi-frame inputs naturally provide richer temporal cues, which can substantially enhance restoration performance. As a promising direction, we plan to extend our simulator to incorporate

temporal information, thereby enabling the exploration of low-cost multi-frame restoration methods. We believe that this line of research will advance turbulence mitigation and push the frontier of robust and efficient restoration under real-world long-range scenarios.

