# OpenReview forum: "Degradation & Restoration: A Low-cost Pipeline for Long-range Single-frame Turbulence Mitigation"
_ICLR.cc/2026/Conference — ICLR 2026 Conference Withdrawn Submission_

### Official Review · Reviewer_jmEe · 2025-10-30

**Soundness:** 2
**Presentation:** 3
**Contribution:** 3
**Rating:** 6
**Confidence:** 5

**Summary:**

This paper presents a single frame turbulence restoration method along with a physics-inspired method for generating diverse turbulence data. On the restoration end, there is some novelty in the finding that the often-used detilt-deblur scheme can be trained completely independently. There is some analysis of the spatially varying blur, though there is some complementary prior work in this space the authors may be unaware of (more later). Finally, the authors visually compare various combinations of simulation modalities and restoration approaches with ablations and some quantitative comparisons.

**Strengths:**

1. The finding that parallely training the de-tilt and de-blur modules retains restoration effectiveness is interesting and, to my knowledge, novel. The training efficiency gained by this is also meaningful.

2. The simulation proposed is well described and reasonably justified. It does not have the same physics-grounding as previous methods, though there is benefit to having the speed and diversity and appears to positively contribute to the image reconstruction.

3. Relative to other single image turbulence restoration methods, this method overall performs better visually. It also performs at or above multi-frame methods, with some caveats I mention in the weaknesses.

**Weaknesses:**

1. There is minimal quantitative comparison between methods on simulated data or real-world data that has some semantic metric (e.g., the text dataset by Mao et al.). Even given the limitations of turbulence comparisons, it should still be offered.

2. The method appears to primarily stabilize the tilt and oversmooth regions (because it is single frame, this is expected and reasonable). The issue is that most of the testing shown is on vehicles with mostly flat details, hence the over-smoothing is not visually severe. Balancing these results with a face or text datasets or even within the dataset from Xu et al. on non-vehicles would be more convincing.

3. Following the previous weakness, I anticipate multi-frame methods could recover details impacted by blur especially for static scenes. Having a sense as to the gap in performance would be convincing.

4. While I regard the mask-then-conv discussion as well-written and beneficial to the paper, there is some prior work in this space. A keyword for this would be “product-convolution” versus “convolution-product”. A few papers as recent as 2024: Lauer, Deconvolution with a spatially variant PSF; Hirsh et al, Efficient filter flow for space-variant multiframe blind deconvolution; Sroubek et al., Decomposition of Space-Variant Blur in Image Deconvolution; Chimitt et al., Scattering and Gather for Spatially Varying Blurs. My intention is not to suggest the authors cite all/any of these papers, rather to present a few key references and make their own assessment.

5. The network itself it not novel, though elements of the training scheme are. I do not consider this a significant weakness, but worth noting.

**Questions:**

Will appreciate the author's thoughts on the weakness commented above.

---

> ### Author Response · Authors · 2025-11-23
> **Rebuttal by Authors**
>
> We strictly appreciate the reviewer's time and the positive assessment of our contributions. We address the specific concerns as follows:
>
> > W1: There is minimal quantitative comparison between methods on simulated data or real-world data that has some semantic metric (e.g., the text dataset by Mao et al.). Even given the limitations of turbulence comparisons, it should still be offered.
>
> A1: We add some quantitative evaluation to response to your concern.
>
> (1) A widly used real-world short-range (300m) text data OCR benchmark [1].
>
> We adopt PaddleOCR [2], one of the most widely used OCR frameworks, and employ its latest model version, PP-OCRv5, rather than outdated OCR models [3, 4] from several years ago. We use only the text detection and text recognition modules in the OCR pipeline, as these provide a clean and interpretable measure of how effectively turbulence mitigation preserves content required for downstream perception. For qualitative evaluation, we follow [1] and employ the AWDR and AD-LCS metrics. The corresponding results are presented below:
>
> | Net              | AWDR$\uparrow$ | AD-LCS$\uparrow$ |
> | ---------------- | -------------- | ---------------- |
> | TurbNet [1]      | 0.966          | 3.744            |
> | PiRN-single [5]  | **0.998**      | 3.290            |
> | AT-DDPM-real [6] | 0.902          | 0.022            |
> | AT-DDPM-syn [6]  | 0.964          | 0.030            |
> | **Ours**         | **0.998**      | **4.146**        |
>
> Even though our simulator is not specifically designed for near-field turbulence, the detilt-deblur network trained on it still achieves **state-of-the-art** performance among single-frame methods. This demonstrates the strong capability and impressive generalization performance of our detilt–deblur network.
>
> (2) A generalization study conducted on simulated data. Due to the physical properties of real long-range turbulence, it is extremely difficult to acquire clean ground truth. To the best of our knowledge, no real-world long-range turbulence dataset provides paired clean and degraded images. Therefore, reference-based IQA metrics such as PSNR, SSIM, and LPIPS cannot be used. Because different works adopt different turbulence simulation strategies, testing existing models on our simulated data, or testing our model on previously released simulated datasets would primarily serve to evaluate the coverage of each simulated distribution and assess the model's ability to generalize across different types of synthetic degradation, rather than to measure absolute turbulence mitigation performance.
>
> Therefore, we train the same model on different simulated datasets and evaluate PSNR on the corresponding test sets. Results are as follow:
>
> (i) evaluate on QuickTurbSim [7].
>
> | input image | trained on ATSyn [8] | trained on ours   |
> | ----------- | -------------------- | ----------------- |
> | 22.81       | 22.66 (-0.15)        | **23.57 (+0.76)** |
>
> (ii) evaluate on ATSyn.
>
> | input image | trained on QuickTurbSim | trained on ours   |
> | ----------- | ----------------------- | ----------------- |
> | 21.36       | 21.12 (-0.24)           | **21.49 (+0.13)** |
>
> (i) and (ii) demonstrate that models trained on our dataset generalize better. This also explains why, in experiment (1), the model trained with our simulated data even generalizes better to short-range real-world turbulence than models trained with short-range simulators.
>
> (iii) evaluate on ours synthetic data.
>
> | input image | trained on QuickTurbSim | trained on ATSyn | trained on ours   |
> | ----------- | ----------------------- | ---------------- | ----------------- |
> | 22.92       | 22.73 (-0.19)           | 22.44 (-0.48)    | **26.41 (+3.49)** |
>
> (iii) demonstrates that models trained on other synthetic datasets fail to generalize effectively to our long-range turbulence degradation. This also indirectly explains why models trained with previous simulators perform worse than ours when applied to real long-range turbulence scenarios.
>
> Note:
>
> - (1) Test set of QuickTurbSim and our simulated data were both regenerated using the same script and identical settings. We are unable to host the original 500+ GB of data online due to storage cost limitations.
> - (2) A recent study [9] shows that nearly all widely used no-reference image quality metrics such as BRISQUE, NIQE, IL-NIQE, PaQ-2-PiQ, etc. are essentially uncorrelated with the turbulence strength $C_n^2$. Therefore, we do not report no-reference IQA metrics in our evaluation since there are not able to reflect the turbulence removal ability.

---

> ### Author Response · Authors · 2025-11-23
> **Rebuttal by Authors (Continue 1)**
>
> > W2: The method appears to primarily stabilize the tilt and oversmooth regions (because it is single frame, this is expected and reasonable). The issue is that most of the testing shown is on vehicles with mostly flat details, hence the over-smoothing is not visually severe. Balancing these results with a face or text datasets or even within the dataset from Xu et al. on non-vehicles would be more convincing.
>
> A2: We are truly grateful for this constructive suggestion, as it helps ensure a more comprehensive and rigorous assessment of our work.    To address the concern regarding dataset bias, we have conducted additional evaluations on the text dataset [1] (detailed in Response A1) and commit to including further scenes from the RLR-AT dataset [10] in the final version.
>
> Regarding the observation on smoothing, we explicitly acknowledge that in scenarios with severe turbulence, our results may indeed appear slightly softer.    This reflects the inherent challenge of information sparsity in single-frame mitigation.    Faced with a delicate trade-off, we prioritized structural fidelity over the risk of generating unrealistic high-frequency artifacts ("hallucinations").    We believe this is a prudent choice, as the "sharpness" observed in some competing methods often stems from uncorrected geometric distortions (i.e., jagged edges caused by pixel displacement) rather than valid detail recovery.    By effectively correcting these distortions, our method yields a more faithfully restored image, a desired visual outcome that we were humbled and pleased to see kindly recognized in the strengths you highlighted.
>
> > W3: Following the previous weakness, I anticipate multi-frame methods could recover details impacted by blur especially for static scenes. Having a sense as to the gap in performance would be convincing.
>
> A3: To quantify this gap, we benchmarked our method against SOTA multi-frame approaches under the same setting as A1.
>
> | Net                 | AWDR$\uparrow$ | AD-LCS$\uparrow$ |
> | ------------------- | -------------- | ---------------- |
> | Ours                | 0.998          | 4.146            |
> | PiRN [5]            | **1.000**      | 3.344            |
> | Turb-Seg-Res [7]    | **1.000**      | 3.818            |
> | DATUM-static [8]    | 0.996          | 4.250            |
> | TMT-static [11]     | **1.000**      | 6.180            |
> | MambaTM-static [12] | **1.000**      | **6.720**        |
>
> Notably, our single-frame method surpasses earlier multi-frame baselines (PiRN, Turb-Seg-Res) in AD-LCS, demonstrating strong feature extraction capabilities. As anticipated, recent SOTA multi-frame methods (e.g., MambaTM) achieve higher scores.
>
> > W4: While I regard the mask-then-conv discussion as well-written and beneficial to the paper, there is some prior work in this space. A keyword for this would be “product-convolution” versus “convolution-product”. A few papers as recent as 2024: Lauer, Deconvolution with a spatially variant PSF; Hirsh et al, Efficient filter flow for space-variant multiframe blind deconvolution; Sroubek et al., Decomposition of Space-Variant Blur in Image Deconvolution; Chimitt et al., Scattering and Gather for Spatially Varying Blurs. My intention is not to suggest the authors cite all/any of these papers, rather to present a few key references and make their own assessment.
>
> A4: Thank you for bringing attention to prior work related to product-convolution formulations. We have reviewed all the listed works. In particular, [13] discusses closely related ideas, and we appreciate this insightful pointer. Its formulation predates and aligns well with the broader topic addressed in our 'mask-then-conv' discussion.
>
> Our observations in the paper were primarily motivated by inspecting the implementations of P2S [14] and QuickTurbSim [7], and we acknowledge that we did not conduct an exhaustive survey of the broader literature on this topic.  We will therefore cite this work and revise the relevant section in the final submission to more accurately position our contribution within the existing research landscape.

---

> ### Author Response · Authors · 2025-11-23
> **Rebuttal by Authors (Continue 2)**
>
> > W5: The network itself it not novel, though elements of the training scheme are. I do not consider this a significant weakness, but worth noting.
>
> A5: We appreciate the reviewer's recognition of the contributions made in our training scheme.    Indeed, the goal of introducing this network is not architectural novelty per se, but to encourage broader engagement from the image restoration community with the turbulence mitigation problem.    By validating our simulator and training strategy on a representative and commonly adopted architecture, we aim to demonstrate that turbulence mitigation can be effectively integrated into existing restoration paradigms.    Furthermore, the acceleration of our simulator and the verification of parallel training feasibility are intended to make future turbulence-mitigation research more accessible and easier to reproduce.    We hope this will support subsequent work, whether in task-specific designs or in general all-in-one restoration models, by enabling long-range turbulence to be incorporated as a meaningful and practically relevant degradation model, ultimately advancing real-world image restoration research.
>
> We once again thank the reviewer for the insightful suggestions and for recognizing our contributions. If our responses adequately address your concerns, we would be sincerely grateful for your positive consideration. Should anything remain unclear, please feel free to let us know, and we would be glad to offer any further clarification.
>
> Reference:
>
> - [1] Single frame atmospheric turbulence mitigation: A benchmark study and a new physics-inspired transformer model. ECCV 2022.
> - [2] https://github.com/PaddlePaddle/PaddleOCR
> - [3] An end-to-end trainable neural network for image-based sequence recognition and its application to scene text recognition. TPAMI 2016.
> - [4] Detecting text in natural image with connectionist text proposal network. ECCV 2016.
> - [5] Physics-driven turbulence image restoration with stochastic refinement. ICCV 2023.
> - [6] At-ddpm: Restoring faces degraded by atmospheric turbulence using denoising diffusion probabilistic models. WACV 2023.
> - [7] Turb-seg-res: A segment-then-restore pipeline for dynamic videos with atmospheric turbulence. CVPR 2024.
> - [8] Spatio-temporal turbulence mitigation: A translational perspective. CVPR 2024.
> - [9] MetaVIn: Meteorological and Visual Integration for Atmospheric Turbulence Strength Estimation. WACV 2025.
> - [10] Long-range turbulence mitigation: a large-scale dataset and a coarse-to-fine framework. ECCV 2024.
> - [11] Imaging through the atmosphere using turbulence mitigation transformer. TCI 2024.
> - [12] Learning Phase Distortion with Selective State Space Models for Video Turbulence Mitigation. CVPR 2025.
> - [13] Scattering and gathering for spatially varying blurs. TSP 2024.
> - [14] Accelerating atmospheric turbulence simulation via learned phase-to-space transform. ICCV 2021.

---

> ### Author Response · Authors · 2025-11-26
>
> Dear Reviewer jmEe,
>
> We hope this message finds you well. Since the discussion phase is approaching its end, we wanted to kindly check whether there are any remaining concerns we could address. Your comments have significantly improved our work, and we genuinely value any additional guidance you might be willing to provide.
> Thank you again for your time and thoughtful input.
>
> Authors of Submission 19612

---

### Official Review · Reviewer_57pP · 2025-10-30

**Soundness:** 2
**Presentation:** 2
**Contribution:** 2
**Rating:** 4
**Confidence:** 3

**Summary:**

This paper addresses long-range atmospheric turbulence mitigation by proposing a physically grounded turbulence degradation simulator. The simulator generates data using tilt and blur. The authors also employ a restoration network to validate the realism of the generated data.

**Strengths:**

1. The paper introduces a physics-based turbulence simulation method and validates its effectiveness.
2. The proposed simulation method achieves relatively high computational efficiency.

**Weaknesses:**

1. The Detilt-then-Deblur network is not fundamentally novel and appears primarily as an engineering contribution.

2. While the simulator is claimed to capture real turbulence statistics, the paper does not sufficiently quantify how well the simulated tilt and blur distributions match real-world data beyond visual comparisons.

3. The lack of rigorous statistical comparison or validation on real-world datasets limits the generalizability of the results.

4. Temporal aspects are underexplored. The simulator only models spatially varying tilt and blur, and single-frame TM cannot inherently capture temporal correlations. The authors’ claim of partial temporal smoothness is a consequence of spatial consistency rather than true temporal learning.

**Questions:**

The key contributions could be stated more explicitly.

Are there statistical metrics, such as the displacement field distribution or PSF energy spectrum, to quantify how well the simulated tilt and blur match real long-range turbulence? Without such metrics, how can the authors ensure the realism of the training data and the generalization of the model?

Can the simulator effectively support restoration of temporally correlated distortions in real dynamic scenes? Have the authors considered extending the simulator to multi-frame or temporal-evolution modeling to improve performance in real-world applications?

---

> ### Author Response · Authors · 2025-11-23
> **Rebuttal by Authors**
>
> We are grateful for your careful evaluation of our work and for the insightful comments you provided. Our detailed responses to the raised concerns are presented as follows:
>
> > W1: The Detilt-then-Deblur network is not fundamentally novel and appears primarily as an engineering contribution.
>
> A1: We appreciate the reviewer's recognition of our detilt-then-deblur network as a solid engineering contribution. We would like to clarify, however, that the network architecture itself is not the primary novelty of our work. Our main contributions lie in: (1) A fast and physically grounded turbulence simulator, which provides realistic supervision for long-range distortions.(2) A two-stage, decoupled training strategy tailored to the distinct components of turbulence degradation.
>
> We also note that, while recent studies have begun exploring all-in-one image restoration models, to the best of our knowledge, turbulence degradation has not yet been incorporated into such unified frameworks.  To further investigate this direction, we evaluate several general-purpose restoration models (CSNet [1], OKNet [2], and ConvIR [3]) and find that even non-specialized architectures for turbulence removal can achieve competitive performance on turbulence mitigation when trained with our simulator. We hope these findings will encourage broader interest from the image restoration community.  Whether considering turbulence removal within task-specific general-purpose restoration models or incorporating it into future all-in-one frameworks, advancing turbulence mitigation remains an important step toward truly comprehensive real-world image restoration under diverse and challenging conditions.

---

> ### Author Response · Authors · 2025-11-23
> **Rebuttal by Authors (Continue 1)**
>
> > W2, W3 & Q1: While the simulator is claimed to capture real turbulence statistics, the paper does not sufficiently quantify how well the simulated tilt and blur distributions match real-world data beyond visual comparisons. The lack of rigorous statistical comparison or validation on real-world datasets limits the generalizability of the results. Are there statistical metrics, such as the displacement field distribution or PSF energy spectrum, to quantify how well the simulated tilt and blur match real long-range turbulence? Without such metrics, how can the authors ensure the realism of the training data and the generalization of the model?
>
> (1) Pixel displacement modeling.
> In the supplementary material of RLR-AT benchmark [4], the authors report statistics measured at 4 km and 6 km, concluding that the displacement at each pixel approximately follows a 2D Gaussian distribution. However, they do not address the spatial correlation inherent to long-range turbulence. In our experiments, applying independent Gaussian perturbations at each pixel leads to patterns resembling random noise rather than physically plausible turbulence. To efficiently generate displacement fields while introducing spatial correlation, we adopt simplex 2D noise. Each pixel's displacement exhibits a bell-shaped distribution similar to a Gaussian, while the inherent spatial smoothness of simplex noise prevents the displacement field from becoming overly random.
>
> (2) Blur modeling.
> Before designing our blur simulation, we conducted preliminary measurements of the RLR-AT benchmark [4] using two established methods [5, 6]. A detailed explanation is provided in our response to reviewer 7eiu.
>
> (3) Generalization to real-world turbulence.
> The visual comparisons presented in the main paper, together with the object-detection evaluations reported in both the main paper and supplementary material, consistently demonstrate that our model exhibits strong generalization capability in real-world long-range turbulence mitigation. Furthermore, we include results on a widely used real-world short-range (300 m) OCR text benchmark [7]. This evaluation, also requested by reviewers 7eiu and jmEe, is broadly recognized as a standard test for short-range turbulence mitigation within the community.
>
> | Net              | AWDR$\uparrow$ | AD-LCS$\uparrow$ |
> | ---------------- | -------------- | ---------------- |
> | TurbNet [7]      | 0.966          | 3.744            |
> | PiRN-single [8]  | **0.998**      | 3.290            |
> | AT-DDPM-real [9] | 0.902          | 0.022            |
> | AT-DDPM-syn [9]  | 0.964          | 0.030            |
> | **Ours**         | **0.998**      | **4.146**        |
>
> Even though our simulator is not specifically designed for near-field turbulence, the detilt-deblur network trained on it still achieves **state-of-the-art** performance among single-frame methods. This demonstrates the strong capability and impressive generalization performance of our detilt–deblur network.
>
> In our response A4 to reviewer Et7z, we compared the detection performance when feeding clean, non-turbulent images into our deblurring and detilting network and observed a significant decrease in accuracy. Together with the results presented in the main paper and the supplementary material, where our model substantially improves detection performance under real long-range turbulence, these findings jointly indicate that training on our simulated dataset effectively reduces the domain gap between the restored outputs and clean images. This, in turn, suggests that our simulator closely captures the degradation characteristics of real atmospheric turbulence.
>
> Due to the character limit, we continue this response below.

---

> ### Author Response · Authors · 2025-11-23
> **Rebuttal by Authors (Continue 2)**
>
> (4) Generalization study conducted on simulated data.
> Due to the physical properties of real long-range turbulence, it is extremely difficult to acquire clean ground truth. To the best of our knowledge, no real-world long-range turbulence dataset provides paired clean and degraded images. Because different works adopt different turbulence simulation strategies, testing existing models on our simulated data, or testing our model on previously released simulated datasets would primarily serve to evaluate the coverage of each simulated distribution and assess the model's ability to generalize across different types of synthetic degradation, rather than to measure absolute turbulence mitigation performance.
>
> Therefore, we train the same model on different simulated datasets and evaluate PSNR on the corresponding test sets. Results are as follow:
>
> (i) evaluate on QuickTurbSim [10].
>
> | input image | trained on ATSyn [11] | trained on ours   |
> | ----------- | --------------------- | ----------------- |
> | 22.81       | 22.66 (-0.15)         | **23.57 (+0.76)** |
>
> (ii) evaluate on ATSyn.
>
> | input image | trained on QuickTurbSim | trained on ours   |
> | ----------- | ----------------------- | ----------------- |
> | 21.36       | 21.12 (-0.24)           | **21.49 (+0.13)** |
>
> (i) and (ii) demonstrate that models trained on our dataset generalize better. This also explains why, in experiment (3), the model trained with our simulated data even generalizes better to short-range real-world turbulence than models trained with short-range simulators.
>
> (iii) evaluate on ours synthetic data.
>
> | input image | trained on QuickTurbSim | trained on ATSyn | trained on ours |
> | ----------- | ----------------------- | ---------------- | --------------- |
> | 22.92       | 22.73 (-0.19)           | 22.44 (-0.48)    | **26.41**       |
>
> (iii) demonstrates that models trained on other synthetic datasets fail to generalize effectively to our long-range turbulence degradation. This also indirectly explains why models trained with previous simulators perform worse than ours when applied to real long-range turbulence scenarios.
>
> Note:
>
> - (1) Test set of QuickTurbSim and our simulated data were both regenerated using the same script and identical settings. We are unable to host the original 500+ GB of data online due to storage cost limitations.
> - (2) A recent study [12] shows that nearly all widely used no-reference image quality metrics such as BRISQUE, NIQE, IL-NIQE, PaQ-2-PiQ, etc. are essentially uncorrelated with the turbulence strength $C_n^2$. Therefore, we do not report no-reference IQA metrics in our evaluation since there are not able to reflect the turbulence removal ability.

---

> ### Author Response · Authors · 2025-11-23
> **Rebuttal by Authors (Continue 3)**
>
> > W4&Q2: Temporal aspects are underexplored. The simulator only models spatially varying tilt and blur, and single-frame TM cannot inherently capture temporal correlations. The authors’ claim of partial temporal smoothness is a consequence of spatial consistency rather than true temporal learning. Can the simulator effectively support restoration of temporally correlated distortions in real dynamic scenes? Have the authors considered extending the simulator to multi-frame or temporal-evolution modeling to improve performance in real-world applications?
>
> We appreciate the reviewer's insightful comments and fully agree that our current experiments demonstrate spatial consistency during restoration, rather than true temporal modeling. Since this work is deliberately focused on single-frame turbulence mitigation, temporal modeling lies outside our intended scope. Thus, its absence should not be viewed as a weakness of the paper.
>
> That said, we would like to offer two clarifications on why single-frame TM remains practically important:
>
> (1) Limitations of existing multi-frame methods.
> State-of-the-art multi-frame approaches such as TMT [13] and DATUM [11] rely on information from adjacent frames. This dependency introduces two practical challenges:
> (i) they require future frames, which is incompatible with real-time or streaming scenarios;
> (ii) they generally incur higher computational latency, making them unsuitable for time-critical applications where delay is unacceptable.
>
> (2) Resolution gap between video and photo capture.
> In most consumer-grade imaging systems, video resolution is significantly lower than photo resolution.
> For example, the Sony A7M4 captures 33 MP photos but only 8.3 MP video. In many real systems, turbulence is barely noticeable at video resolution but becomes strongly visible in high-resolution photos. This difference further motivates the need for high-fidelity single-frame restoration, even when multi-frame sequences are available.
>
> Even so, we carefully considered the reviewer's suggestion regarding temporal modeling and provide here a preliminary extension idea inspired by real-world turbulence behavior:
>
> (1) Physical motivation: horizontal advection.
> From our analysis of the RLR-AT benchmark [14], we observed that most turbulence patterns exhibit horizontally advected distortions, which originate from horizontal airflows near the ground. Based on this physical property, both tilt and blur simulation can be extended by replacing fixed sampling grids with horizontally sliding windows.
> The sliding motion naturally introduces temporal evolution, while the window velocity controls the wind speed, and additional noise injected during sliding can model diffusion strength.
>
> (2) Engineering feasibility: avoiding OOM.
> When long sliding windows cause memory overflow, the temporal grid can be implemented using blended masks and window stitching, ensuring the effective window length does not exceed roughly three times the original size. This makes the temporal extension computationally feasible.
>
> We hope this preliminary direction addresses your concern and demonstrates that our simulator can be naturally extended toward temporal-evolution modeling if needed.
>
> If our responses adequately address your concerns, we would be sincerely grateful for your positive consideration.   Should anything remain unclear, please do not hesitate to let us know.  we would be happy to provide any further clarification.   Thank you again for the time and thoughtful attention you have given to our work.
>
> Reference
>
> - [1] Hybrid frequency modulation network for image restoration. IJCAI 2024.
> - [2] Omni-kernel network for image restoration. AAAI 2024.
> - [3] Revitalizing convolutional network for image restoration. TPAMI 2024.
> - [4] Turb-seg-res: A segment-then-restore pipeline for dynamic videos with atmospheric turbulence. CVPR 2024.
> - [5] An end-to-end trainable neural network for image-based sequence recognition and its application to scene text recognition. TPAMI 2016.
> - [6] Detecting text in natural image with connectionist text proposal network. ECCV 2016.
> - [7] Single frame atmospheric turbulence mitigation: A benchmark study and a new physics-inspired transformer model. ECCV 2022.
> - [8] Physics-driven turbulence image restoration with stochastic refinement." ICCV 2023.
> - [9] At-ddpm: Restoring faces degraded by atmospheric turbulence using denoising diffusion probabilistic models. WACV 2023.
> - [10] Turb-seg-res: A segment-then-restore pipeline for dynamic videos with atmospheric turbulence. CVPR 2024.
> - [11] Spatio-temporal turbulence mitigation: A translational perspective. CVPR 2024.
> - [12] MetaVIn: Meteorological and Visual Integration for Atmospheric Turbulence Strength Estimation. WACV 2025.
> - [13] Imaging through the atmosphere using turbulence mitigation transformer. TCI 2024.
> - [14] Long-range turbulence mitigation: a large-scale dataset and a coarse-to-fine framework. ECCV 2024.

---

> ### Author Response · Authors · 2025-11-26
>
> Dear Reviewer 57pP,
>
> We hope everything is going well. As the discussion window is closing soon, we wanted to humbly check whether there is anything more we can clarify. Your suggestions have been invaluable to us, and we would be more than happy to elaborate on any point you feel needs further attention.
> Thank you so much for your effort and for helping us strengthen our paper.
>
> Authors of Submission 19612

---

### Official Review · Reviewer_Et7z · 2025-10-31

**Soundness:** 2
**Presentation:** 3
**Contribution:** 1
**Rating:** 2
**Confidence:** 3

**Summary:**

The paper proposes a sophisticated turbulence simulation pipeline, and trains a turbulence removal model based on it. The realistic simulation pipeline also enables a parallel training of the detilt-then-deblur network, to better disentangle the two degradations.

**Strengths:**

1. Low computatoinal cost.
2. Turbulence simulator. This could benefit the field of turbulence removal, if the authors were to open-source the project.

**Weaknesses:**

1. No quantitative evaluation. The shown visual comparisons do not necessarily lead to the conclusion of a SOTA performance. Sometimes the proposed method suffers from a loss of details and over-smoothing. Authors should report the PSNR, SSIM, and LPIPS performances on an established dataset.

2. Very few examples. I thought the supplementary may contain maybe ten cases or so, since the paper does not contain many cases. However I only found 2, which is insufficient.

3. The paper puts a lot of emphasis on the tilt and blur simulation, and the deblurring and detilting do not have much novelty. This may be a little different from this conference, which still focuses more on learning than simulation. Perhaps a graphics-related conference may be better for this topic?

**Questions:**

1. Would applying the deblurring and detilting network deteriorate the YOLO detection performance, given a clean image as input?

---

> ### Author Response · Authors · 2025-11-23
> **Rebuttal by Authors**
>
> We sincerely thank you for reviewing our paper and providing us valuable feedback. We have addressed your concerns as below.
>
> > W1-1: No quantitative evaluation. The shown visual comparisons do not necessarily lead to the conclusion of a SOTA performance. Authors should report the PSNR, SSIM, and LPIPS performances on an established dataset.
>
> A1-1: We add some quantitative evaluation to response to your concern.
>
> (1) A widly used real-world short-range (300m) text data OCR benchmark [1]. This evaluation, also requested by reviewer 7eiu and reviewer jmEe, is widely recognized as a standard test for short-range turbulence mitigation in the community.
>
> We adopt PaddleOCR [2], one of the most widely used OCR frameworks, and employ its latest model version, PP-OCRv5, rather than outdated OCR models [3, 4] from several years ago. We use only the text detection and text recognition modules in the OCR pipeline, as these provide a clean and interpretable measure of how effectively turbulence mitigation preserves content required for downstream perception. For qualitative evaluation, we follow [1] and employ the AWDR and AD-LCS metrics. The corresponding results are presented below:
>
> | Net              | AWDR$\uparrow$ | AD-LCS$\uparrow$ |
> | ---------------- | -------------- | ---------------- |
> | TurbNet [1]      | 0.966          | 3.744            |
> | PiRN-single [5]  | **0.998**      | 3.290            |
> | AT-DDPM-real [6] | 0.902          | 0.022            |
> | AT-DDPM-syn [6]  | 0.964          | 0.030            |
> | **Ours**         | **0.998**      | **4.146**        |
>
> Even though our simulator is not specifically designed for near-field turbulence, the detilt-deblur network trained on it still achieves **state-of-the-art** performance among single-frame methods. This demonstrates the strong capability and impressive generalization performance of our detilt–deblur network.
>
> (2) A generalization study conducted on simulated data. Due to the physical properties of real long-range turbulence, it is extremely difficult to acquire clean ground truth. To the best of our knowledge, no real-world long-range turbulence dataset provides paired clean and degraded images. Therefore, reference-based IQA metrics such as PSNR, SSIM, and LPIPS **cannot be used**. Because different works adopt different turbulence simulation strategies, testing existing models on our simulated data, or testing our model on previously released simulated datasets would primarily serve to evaluate the coverage of each simulated distribution and assess the model’s ability to generalize across different types of synthetic degradation, rather than to measure absolute turbulence mitigation performance.
>
> Therefore, we train the same model on different simulated datasets and evaluate PSNR on the corresponding test sets. Results are as follow:
>
> (i) evaluate on QuickTurbSim [7].
>
> | input image | trained on ATSyn [8] | trained on ours   |
> | ----------- | -------------------- | ----------------- |
> | 22.81       | 22.66 (-0.15)        | **23.57 (+0.76)** |
>
> (ii) evaluate on ATSyn.
>
> | input image | trained on QuickTurbSim | trained on ours   |
> | ----------- | ----------------------- | ----------------- |
> | 21.36       | 21.12 (-0.24)           | **21.49 (+0.13)** |
>
> (i) and (ii) demonstrate that models trained on our dataset generalize better. This also explains why, in experiment (1), the model trained with our simulated data even generalizes better to short-range real-world turbulence than models trained with short-range simulators.
>
> (iii) evaluate on ours synthetic data.
>
> | input image | trained on QuickTurbSim | trained on ATSyn | trained on ours   |
> | ----------- | ----------------------- | ---------------- | ----------------- |
> | 22.92       | 22.73 (-0.19)           | 22.44 (-0.48)    | **26.41 (+3.49)** |
>
> (iii) demonstrates that models trained on other synthetic datasets fail to generalize effectively to our long-range turbulence degradation. This also indirectly explains why models trained with previous simulators perform worse than ours when applied to real long-range turbulence scenarios.
>
> Note:
>
> - (1) Test set of QuickTurbSim and our simulated data were both regenerated using the same script and identical settings. We are unable to host the original 500+ GB of data online due to storage cost limitations.
> - (2) A recent study [9] shows that nearly all widely used no-reference image quality metrics such as BRISQUE, NIQE, IL-NIQE, PaQ-2-PiQ, etc. are essentially uncorrelated with the turbulence strength $C_n^2$. Therefore, we do not report no-reference IQA metrics in our evaluation since there are not able to reflect the turbulence removal ability.

---

> ### Author Response · Authors · 2025-11-23
> **Rebuttal by Authors (Continue 1)**
>
> > W1-2: Sometimes the proposed method suffers from a loss of details and over-smoothing.
>
> A1-2: We appreciate the reviewer's keen observation.  We acknowledge that in certain scenarios with severe turbulence, our method may produce results that appear slightly smoothed.  This is primarily due to the inherent information sparsity in single-frame turbulence mitigation.  With only single-frame input, there is a delicate trade-off between hallucinating unrealistic high-frequency artifacts and producing a clean, albeit smoother, reconstruction. We prioritized structural fidelity over generating potential artifacts.  It is worth noting that while some competing methods may appear "sharper," this sharpness often stems from uncorrected geometric distortions (i.e., high-frequency edges caused by pixel displacement) rather than recovered true details.  Our method effectively corrects these distortions, leading to a more faithful restoration.  As Reviewer jmEe noted, "Relative to other single image turbulence restoration methods, this method overall performs better visually," confirming that our trade-off results in superior overall perceptual quality.
>
> > W2: Very few examples. I thought the supplementary may contain maybe ten cases or so, since the paper does not contain many cases. However I only found 2, which is insufficient.
>
> A2: We appreciate the reviewer's interest in seeing a broader range of scenarios. In the final version, we will include a more diverse set of visualizations covering a wider variety of scene types, turbulence patterns, and restoration outcomes to present a clearer and more comprehensive picture of the method's behavior.
>
> > W3: The paper puts a lot of emphasis on the tilt and blur simulation, and the deblurring and detilting do not have much novelty. This may be a little different from this conference, which still focuses more on learning than simulation. Perhaps a graphics-related conference may be better for this topic?
>
> A3: We appreciate the reviewer's recognition of our contributions on the simulation side. We believe our work is well aligned with the scope and interests of ICLR, for several reasons:
>
> (1) Stronger modeling leads to stronger learning. As we stated in A1, Our work introduces a new turbulence simulator specifically designed to address a critical limitation of previous simulators that models trained on them fail to recover object shapes under long-range turbulence conditions. This strongly suggests that either (i) current models are unable to learn the degradation characteristics present in the synthetic data, or (ii) the simulated turbulence does not faithfully match real long-range degradation. Our simulator is introduced precisely to provide a learning signal that is statistically aligned with real turbulence, enabling our detilt-deblur network to learn behaviors that were previously inaccessible.
>
> (2) Our two-stage separated training framework also introduces innovation on the learning side. Motivated by the physics of propagation, this design not only accelerates training but also allows each stage to specialize in a distinct degradation component, resulting in markedly improved performance on both synthetic and real-world data.
>
> (3) ICLR explicitly welcomes application-driven contributions. According to the description on official website [10], the conference highlights advances in deep learning across both foundational methods and important application domains, including machine vision:
>
> > ICLR is globally renowned for presenting and publishing cutting-edge research on all aspects of deep learning used in the fields of artificial intelligence, statistics and data science, as well as important application areas such as machine vision ...
>
> > A non-exhaustive list of relevant topics explored at the conference include:
> >
> > - applications in audio, speech, robotics, neuroscience,  biology, or any other field
>
> Our paper aims to advance single-frame turbulence mitigation, which is fundamentally a computer vision application. We also submitted the paper under the primary area "applications to computer vision, audio, language, and other modalities," which is fully consistent with the conference's guidelines and scope.

---

> ### Author Response · Authors · 2025-11-23
> **Rebuttal by Authors (Continue 2)**
>
> > Q1: Would applying the deblurring and detilting network deteriorate the YOLO detection performance, given a clean image as input?
>
> A4: We appreciate you bringing up this idea, which we had not carefully examined before. We conducted the requested evaluation using the same YOLO11x [11] detector as in our main paper. The results on MS COCO val2017 [12] are as follows:
>
> | Metric   | using our network | clean image |
> | -------- | ----------------- | ----------- |
> | mAP50-95 | 0.3720            | 0.5485      |
> | AP50     | 0.5266            | 0.7135      |
> | AP75     | 0.3974            | 0.5974      |
> | mPrec    | 0.6593            | 0.7368      |
> | mRecall  | 0.4779            | 0.6588      |
>
> The detection accuracy on clean images decreases after passing through our network. This behavior **is expected**, because our method is specifically designed for long-range turbulence mitigation, rather than for enhancing detection performance on normal, non-turbulent images. Our training data do not include turbulence-free images, and the model is therefore not optimized for identity preservation on natural clean inputs. Consequently, limited generalization to such out-of-domain cases is reasonable and acceptable.
>
> At the same time, this experiment provides an important insight: our network does not universally boost downstream performance for arbitrary images. Instead, it improves detection performance only under long-range turbulence conditions, which is precisely the regime our simulator and restoration network are designed for. This observation further supports that the **model has learned the statistical characteristics of real turbulence** rather than acting as a generic enhancer on ordinary scenes.
>
> If one wishes to improve detection performance across all scenarios, a practical solution would be to incorporate a lightweight pre-classification module that determines whether turbulence is present before invoking our restoration network. We mention this only as a potential direction beyond the scope of this paper.
>
> If our responses sufficiently address the concerns you raised, we would greatly appreciate your favorable consideration. If any points remain unclear, please feel free to let us know. We are glad to provide further clarification. Thank you again for the time and care you devoted to reviewing our work.
>
> Reference
>
> - [1] Single frame atmospheric turbulence mitigation: A benchmark study and a new physics-inspired transformer model. ECCV 2022.
> - [2] https://github.com/PaddlePaddle/PaddleOCR
> - [3] An end-to-end trainable neural network for image-based sequence recognition and its application to scene text recognition. TPAMI 2016.
> - [4] Detecting text in natural image with connectionist text proposal network. ECCV 2016.
> - [5] Physics-driven turbulence image restoration with stochastic refinement. ICCV 2023.
> - [6] At-ddpm: Restoring faces degraded by atmospheric turbulence using denoising diffusion probabilistic models. WACV 2023.
> - [7] Turb-seg-res: A segment-then-restore pipeline for dynamic videos with atmospheric turbulence. CVPR 2024.
> - [8] Spatio-temporal turbulence mitigation: A translational perspective. CVPR 2024.
> - [9] MetaVIn: Meteorological and Visual Integration for Atmospheric Turbulence Strength Estimation. WACV 2025.
> - [10] https://iclr.cc/Conferences/2026
> - [11] https://github.com/ultralytics/ultralytics
> - [12] Microsoft coco: Common objects in context. ECCV 2014.

---

> ### Author Response · Authors · 2025-11-26
>
> Dear Reviewer Et7z,
>
> We hope this message finds you well. If there are still any remaining concerns, please kindly let us know. We will spare no effort to address them thoroughly. We sincerely appreciate your valuable comments, constructive suggestions, and the time and effort you have devoted to reviewing our work. Your guidance is truly invaluable to us.
>
> Authors of Submission 19612

---

> > ### Comment · Reviewer_Et7z · 2025-11-28
> > **Response to authors**
> >
> > That is some pretty lengthy rebuttal, and I appeciate the authors for their efforts. Most of my concerns are addressed, except for the metrics I mentioned. Is there any difficulty in measuring those? I thought they are critical metrics in evaluating reconstruction performance.
> >
> > Nevertheless, I would raise my rating, as a part of my concerns are addressed.

---

> > > ### Author Response · Authors · 2025-11-28
> > >
> > > Dear reviewer Et7z,
> > >
> > > Thank you sincerely for your thoughtful reply and for your kind recognition of our work. We completely agree that PSNR, SSIM, and LPIPS are important and widely used restoration metrics. For long-range turbulence, however, to our knowledge there is no real-world dataset with paired clean ground truth, which makes these reference-based metrics unfortunately not applicable. Following the helpful suggestions from reviewers 7eiu and jmEe, we adopted widely used text-based evaluation for real-world assessment (see Response A1-1(1) ). We truly appreciate your valuable suggestions and your supportive comments.
> > >
> > > Authors of Submission 19612

---

### Official Review · Reviewer_7eiu · 2025-10-31

**Soundness:** 4
**Presentation:** 4
**Contribution:** 4
**Rating:** 4
**Confidence:** 4

**Summary:**

This paper proposes a new simulator for turbulence data, with more realistic and detailed tilts and efficient spatially varying blur. as well as a new and more efficient training scheme for a turbulence mitigation architecture. The authors show that architectures trained on their simulated achieve better performance than when trained on that of other simulations.

**Strengths:**

The improvements in the simulation process are well-motivated and sensical. The tilt model attempts to capture the multiscale features seen in turbulence using various amounts of simplex noise, as well as the amount of warping due to turbulence with random warp iterations. Furthermore, the changes to make the spatially varying blur implementable efficiently on GPU, namely, their mask-the-conv scheme, are well explained.

As for evaluation, the two-stage method seems to perform comparable to the state of the art while taking much less time.

**Weaknesses:**

I'm not sure what the authors mean when they say "the displacement (tilt) fields produced by current tilt simulators deviate substantially
from the spatial statistics of real long-range turbulence" in the introduction. While the limitations in how prior methods simulate blur is discussed in the related works section, I could not find much said about the tilt.

As for the blur, the authors say they develop their random kernel generator based on measured PSFs, but I could not find any detail on where they come from, whether from the literature or the authors' own experiments.

The qualitative results are not clear and the examples could be better chosen. For example, in the turbulence literature it is common to see if models can mitigate turbulence that obscures text in images [1].

[1] https://openaccess.thecvf.com/content/CVPR2025/html/Zhang_Learning_Phase_Distortion_with_Selective_State_Space_Models_for_Video_CVPR_2025_paper.html

**Questions:**

- How is the current simulation better at modeling tilt than prior simulations?
- Where do the measured PSFs that the authors base their blur simulation on come from?

---

> ### Author Response · Authors · 2025-11-23
> **Rebuttal by authors**
>
> We thank you for carefully reading our paper and for providing thoughtful and constructive feedback. Our responses to the identified weaknesses and questions are as follows:
>
> > Q1&W1: How is the current simulation better at modeling tilt than prior simulations? I'm not sure what the authors mean when they say "the displacement (tilt) fields produced by current tilt simulators deviate substantially from the spatial statistics of real long-range turbulence" in the introduction. While the limitations in how prior methods simulate blur is discussed in the related works section, I could not find much said about the tilt.
>
> A1: We appreciate you for pointing out that our explanation of the tilt modeling limitations in prior simulators was not sufficiently clear. We clarify below. The baseline QuickTurbSim [1] relies on a single-scale simplex-noise field as the pixel-displacement map. This forces all simulated tilts within a patch to share one spatial scale, which contradicts the multi-scale, spatially irregular statistics of real long-range turbulence. In addition, QuickTurbSim applies only a single warp, preventing it from reflecting the fact that long-range propagation involves multiple phase-screen refractions that progressively alter the displacement direction.
>
> In section 3.2, we replace the single-scale displacement with a multi-scale combination of 2D simplex noises, enabling each warp to introduce tilts of different spatial frequencies. This better matches the stochastic spatial structure of real turbulence. Moreover, we apply multiple sequential warp operations, mimicking the progressive, distance-dependent displacement observed in long-range propagation. These two modifications significantly improve the fidelity of simulated tilt fields.
>
> Unlike P2S-type physical simulators P2S [2] and ATSyn [3], which require expensive computation of $S_{half}$ for each propagation step, our multi-warp displacement approach is orders of magnitude faster while retaining the essential spatial statistics of real turbulence. This improved tilt modeling also translates into better real-world performance: models trained with our simulator achieve more accurate shape recovery, also results as higher performance in object detection.
>
> > Q2&W2: Where do the measured PSFs that the authors base their blur simulation on come from? As for the blur, the authors say they develop their random kernel generator based on measured PSFs, but I could not find any detail on where they come from, whether from the literature or the authors' own experiments.
>
> A2: We first estimated the degradation kernels on RLR-AT benchmark [4] using MANet [5] and MLMC [6], two representative kernel-based blind super-resolution approaches. There are two main reasons for this choice. First, our survey did not reveal any open-source PSF estimators tailored for long-range atmospheric turbulence. Second, the degradation kernels estimated by blind super-resolution methods effectively approximate the underlying blur operator (i.e., the PSF). This is well aligned with our setting: since we train the detilt and deblur stages separately, the deblur stage can be considered as blind image restoration, which is conceptually close to blind super-resolution. Using MANet/MLMC thus provides kernel estimates that are more compatible with our network architecture and training workflow.
>
> It is important to emphasize that the shape of our designed random kernel is intentionally not identical to the PSFs estimated in [1, 2]. Those works report PSFs that exhibit an almost perfectly "large center, small periphery" pattern, essentially resembling a stretched Gaussian kernel. However, convolving with such kernels results in severe loss of high-frequency content, and in our experiments, none of the representative restoration networks (CSNet [7], OKNet [8], and ConvIR [9]) were able to recover satisfactory details when trained with these highly smooth PSFs.
>
> Motivated by this observation, our kernel design preserves the general large-center structure but injects controlled random perturbations, ensuring that part of the high-frequency information is retained in the degraded images.  This leads to substantially better deblurring performance in practice, as the degradation remains physically plausible while still providing a learnable signal for reconstruction networks.

---

> ### Author Response · Authors · 2025-11-23
> **Rebuttal by Authors (Continue)**
>
> > W3: The qualitative results are not clear and the examples could be better chosen. For example, in the turbulence literature it is common to see if models can mitigate turbulence that obscures text in images.
>
> A3: Our simulator is designed for long-range turbulence mitigation, whereas the text dataset [10] is collected at approximately 300 meters, which corresponds to a relatively short propagation distance in turbulence studies. This mismatch is the main reason why we did not include comparisons on this dataset in the paper.
>
> Nevertheless, to address your concern, we conducted inference on this dataset. We adopt PaddleOCR [11], one of the most widely used OCR frameworks, and employ its latest model version, PP-OCRv5, rather than outdated OCR models [12, 13] from several years ago. We use only the text detection and text recognition modules in the OCR pipeline, as these provide a clean and interpretable measure of how effectively turbulence mitigation preserves content required for downstream perception. For qualitative evaluation, we follow [10] and employ the AWDR and AD-LCS metrics. The corresponding results are presented below:
>
> | Net               | AWDR$\uparrow$ | AD-LCS$\uparrow$ |
> | ----------------- | -------------- | ---------------- |
> | TurbNet [10]      | 0.966          | 3.744            |
> | PiRN-single [14]  | **0.998**      | 3.290            |
> | AT-DDPM-real [15] | 0.902          | 0.022            |
> | AT-DDPM-syn [15]  | 0.964          | 0.030            |
> | **Ours**          | **0.998**      | **4.146**        |
>
> Even though our simulator is not specifically designed for short-range turbulence, the detilt-deblur network trained on it still achieves **state-of-the-art** performance among single-frame methods.
>
> We also provide a generalization study on simulated data to demonstrate the strong generalization ability of models trained on our dataset. See our reply A1-1 (2) to reviewer Et7z.
>
> If our responses adequately address the weaknesses and questions you raised, we would be grateful for your positive consideration. If not, please let us know which parts remain unclear. We sincerely thank you again for the careful review of our work!
>
> Reference
>
> - [1] Turb-seg-res: A segment-then-restore pipeline for dynamic videos with atmospheric turbulence. CVPR 2024.
> - [2] Accelerating atmospheric turbulence simulation via learned phase-to-space transform. ICCV 2021.
> - [3] Spatio-temporal turbulence mitigation: A translational perspective. CVPR 2024.
> - [4] Long-range turbulence mitigation: a large-scale dataset and a coarse-to-fine framework. ECCV 2024.
> - [5] Mutual affine network for spatially variant kernel estimation in blind image super-resolution. ICCV 2021.
> - [6] Blind super-resolution via meta-learning and Markov chain Monte Carlo simulation. TPAMI 2024.
> - [7] Hybrid frequency modulation network for image restoration. IJCAI 2024.
> - [8] Omni-kernel network for image restoration. AAAI 2024.
> - [9] Revitalizing convolutional network for image restoration. TPAMI 2024.
> - [10] Single frame atmospheric turbulence mitigation: A benchmark study and a new physics-inspired transformer model. ECCV 2022.
> - [11] https://github.com/PaddlePaddle/PaddleOCR
> - [12] An end-to-end trainable neural network for image-based sequence recognition and its application to scene text recognition. TPAMI 2016.
> - [13] Detecting text in natural image with connectionist text proposal network. ECCV 2016.
> - [14] Physics-driven turbulence image restoration with stochastic refinement. ICCV 2023.
> - [15] At-ddpm: Restoring faces degraded by atmospheric turbulence using denoising diffusion probabilistic models. WACV 2023.

---

> ### Author Response · Authors · 2025-11-26
>
> Dear Reviewer 7eiu,
>
> We hope this message finds you well. Since the discussion phase is approaching its end, we wanted to kindly check whether there are any remaining concerns we could address. Your comments have significantly improved our work, and we genuinely value any additional guidance you might be willing to provide. Thank you again for your time and thoughtful input.
>
> Authors of Submission 19612

---

### Author Response · Authors · 2025-12-03
**Author Final Remarks**

Dear AC,

Thank you very much for taking the time to carefully review our submission and discussion.   During the discussion phaes, only Reviewer Et7z responded promptly and indicated that they would revise their score.   Unfortunately, the remaining reviewers did not reply before the discussion period was closed under the updated review policy.   We were disappointed to learn that the reviewers were no longer able to provide additional feedback, as we put substantial effort into addressing every comment and were hoping for further clarification to help us improve the paper.

To facilitate your reading and to enable an accurate assessment of our work, we summarize below:

### 1. Main contribution of this paper.

Our work is motivated by the observation that (i) traditional multi-frame fusion approaches fail in dynamic scenes (ii) although several deep-learning–based methods have demonstrated promising results on short-range turbulence, our real-world experiments reveal that these methods struggle to recover long-range turbulence, which we attribute primarily to a significant distribution mismatch between existing synthetic training data and real long-range turbulence.           (iii) We further found that existing turbulence simulators are computationally slow, making large-scale data generation impractical.           To address above concerns, we propose a fast yet effective turbulence simulator for DL method, composed of a tilt simulator based on multi-step warping and multi-scale simplex noise, and a blur simulator based on gradual transition mask and random kernels.

(iv) during reproduction of prior methods, we observed that existing models require extremely heavy training, often involving many A100 GPUs for several weeks.           (v) many of these models also rely on specialized physics-driven modules, making them incompatible with general image restoration frameworks.           To overcome this two question, we develop a detilt-then-deblur architecture that supports parallel training of two-stage, significantly reducing the time of training.           Our design also allows standard image restoration networks to be used as backbones, without requiring additional physics modules.
Finally, our real-world evaluation demonstrates that the proposed single-frame restoration method achieves state-of-the-art performance on both static and dynamic long-range turbulence scenes, outperforming several multi-frame models across multiple scenarios.

### 2. Additions made during the discussion phase.
To address the reviewers’ common request for more qualitative experiments, we added results on the text-based benchmark recommended independently by Reviewer 7eiu and Reviewer jmEe.       Even though our simulator does not explicitly model short-range turbulence, our method still achieves state-of-the-art single-frame turbulence mitigation (TM) performance on this benchmark.       This further supports the strong generalization ability of our network to real-world scenarios.

We fully understand that the reviewers wished to see more comprehensive evidence to better assess our contributions.       Therefore, we included additional visual comparisons in both the main paper and Appendix C. Since ground-truth clean images for long-range turbulence are extremely difficult to acquire—and no real datasets provide paired clean/degraded images—we remain limited to visual evaluations for real-world experiments.

We also added experiments evaluating cross-dataset generalization among different synthetic turbulence datasets.       These results show that networks trained on other synthetic datasets fail to generalize well to our simulated data, whereas models trained on our dataset generalize effectively to other simulators.       Additionally, we thank Reviewer jmEe for pointing out the missing discussion on related work of 'mask-then-conv';       we have now incorporated the related analysis into related section.

### 3. Other clarifications provided during the discussion but not incorporated into the manuscript.
For this part, we respectfully direct you to the full discussion thread.       We provided detailed answers to every reviewer question.       We hope that you will find our responses thorough and satisfactory.

Once again, we sincerely appreciate your careful and detailed assessment of our submission.

Best regards,

Authors of submission 19612

---

### Note · Authors · 2026-01-27

I have read and agree with the venue's withdrawal policy on behalf of myself and my co-authors.

---

### Meta-Review · Area_Chair_hbag · 2025-12-23

**Summary:**

All reviewers raised major concerns regarding the simulation methodology proposed in this paper and the corresponding evaluation protocols. The literature on turbulence simulation and mitigation is extensive; however, the authors fail to present adequate comparison metrics or situate their approach appropriately within existing work.

**Reviewer Concerns:**

The authors have attempted to address some of the concerns by conducting additional experiments on object detection and turbulence mitigation. However, appropriate evaluation metrics were not used, and the resulting evidence remains inconclusive.

**Reviewer Scores:**

A major concern relates to the evaluation of the proposed simulation method. Simulation is valuable insofar as it meaningfully contributes to understanding or solving real-world problems; without convincing evidence of this, it is difficult to assess the practical significance of the proposed approach.

The authors argue that metrics such as PSNR and SSIM are not appropriate due to the lack of ground truth. However, one reasonable alternative would be to apply the proposed simulation to clean images or videos and treat these as synthetic ground truth for quantitative evaluation. In this context, the authors’ dismissal of standard evaluation metrics is somewhat puzzling and warrants further clarification.

In addition, comparisons with more recent turbulence mitigation methods are missing. For example, several recent works by S. Chan on turbulence simulation and mitigation would provide important context and a more comprehensive evaluation of the proposed method.

---

### Decision · Program_Chairs · 2026-01-26

Reject